# Improved Learning via k-DTW: A Novel Dissimilarity Measure for Curves

**Amer Krivošija** [1]   **Alexander Munteanu** [1 2]   **André Nusser** [3]   **Chris Schwiegelshohn** [4]

## Abstract

This paper introduces $k$-Dynamic Time Warping ($k$-DTW), a novel dissimilarity measure for polygonal curves. $k$-DTW has stronger metric properties than Dynamic Time Warping (DTW) and is more robust to outliers than the Fréchet distance, which are the two gold standards of dissimilarity measures for polygonal curves. We show interesting properties of $k$-DTW and give an exact algorithm as well as a $(1 + \varepsilon)$-approximation algorithm for $k$-DTW by a parametric search for the $k$-th largest matched distance. We prove the first dimension-free learning bounds for curves and further learning theoretic results. $k$-DTW not only admits smaller sample size than DTW for the problem of learning the median of curves, where some factors depending on the curves' complexity $m$ are replaced by $k$, but we also show a surprising separation on the associated Rademacher and Gaussian complexities: $k$-DTW admits strictly smaller bounds than DTW, by a factor $\tilde{\Omega}(\sqrt{m})$ when $k \ll m$. We complement our theoretical findings with an experimental illustration of the benefits of using $k$-DTW for clustering and nearest neighbor classification.

## 1. Introduction

Handwriting, panel data, time series, sensor-generated trajectories, and many more types of data are instances of polygonal curves in Euclidean space. These curves are input for learning processes, both supervised and unsupervised. To compare and quantify the distance of two curves, a suitable dissimilarity measure is needed that reflects the fact that similar curves that represent, for instance, the same handwritten characters, can be sampled differently and can therefore look very different at the data level, even though they are similar in content and visual appearance.

Dissimilarity measures that involve transformations of one curve into another are therefore studied across the literature. Prominent examples are the Fréchet distance, arguably the most popular measure in computational geometry, or the Dynamic Time Warping (DTW) distance, which is often used in the context of machine learning. Both can be computed in near-quadratic time in terms of the number of vertices representing the two curves (Alt & Godau, 1995; Eiter & Mannila, 1994; Vintsyuk, 1968). There is also evidence that the Fréchet distance as well as DTW cannot be computed in strongly subquadratic time, based on conditional lower bounds that rely on widely accepted complexity theoretic conjectures (Bringmann, 2014; Bringmann & Mulzer, 2016; Abboud et al., 2015; Bringmann & Künnemann, 2015; Buchin et al., 2019c). Hence, it is widely believed that only small polylogarithmic improvements can be achieved (Agarwal et al., 2014; Gold & Sharir, 2018) over the quadratic complexity of the natural dynamic programming approach.

Both dissimilarity measures have their caveats: while the Fréchet distance is a metric, it is however very sensitive to outlier vertices, to such a degree that the existence of a single outlier completely determines the distance. DTW is much less outlier-sensitive, but it is not a metric since it does not satisfy the triangle inequality, which can be violated by a large factor that depends on the length of the curve. These problems can unfavorably affect the outcome when Fréchet or DTW are applied for clustering or classification.

### 1.1. Our Contributions

We propose a novel dissimilarity measure $k$-DTW that provides the "best of both worlds", while generalizing and interpolating between the Fréchet, and the DTW distance. $k$-DTW considers only a small subset of size $k$ comprising the most important part of the transformation between two curves, and ignores the remaining transformation that matches only small deviations, which often correspond to noise. We summarize our main contributions as follows:

1) We prove that $k$-DTW provides a strengthened triangle inequality compared to DTW and it is thus closer to a proper metric, while retaining some robustness of DTW.

[1]TU Dortmund, Germany [2]University of Cologne, Germany [3]Université Côte d'Azur, CNRS, Inria, France [4]Aarhus University, Denmark. Correspondence to: A. Krivošija <amer.krivosija@tu-dortmund.de>, A. Munteanu <alexander.munteanu@tu-dortmund.de>, A. Nusser <andre.nusser@cnrs.fr>, C. Schwiegelshohn <cschwiegelshohn@gmail.com>.

*Proceedings of the $42^{nd}$ International Conference on Machine Learning*, Vancouver, Canada. PMLR 267, 2025. Copyright 2025 by the author(s).

2) We provide an exact algorithm as well as a $(1 + \varepsilon)$-approximation algorithm for $k$-DTW using a parametric search for the $k$-th largest matched distance with standard DTW on modified distance matrices as a subroutine.
3) We prove the first dimension-free learning bounds for clustering under $k$-DTW (including DTW and Fréchet) and a separation result showing that $k$-DTW has strictly smaller Rademacher and Gaussian complexity than DTW for clustering curves.
4) We experimentally show the benefits of $k$-DTW over the Fréchet distance and DTW for clustering and nearest neighbor classification of synthetic and real world data.

### 1.2. Other Related Work

Many dissimilarity measures were introduced to alleviate the drawbacks of the Fréchet distance and DTW. Examples include the weak Fréchet distance (Alt & Godau, 1995) and Continuous Dynamic Time Warping (CDTW) (Buchin, 2007; Buchin et al., 2022). Also, clustering under these measures was considered with different degrees of success. While center based clustering of curves is NP-hard in most cases (Driemel et al., 2016; Buchin et al., 2019a; 2020; Bulteau et al., 2020), there exist practical approaches to cluster curves under the Fréchet distance (Buchin et al., 2019b) and CDTW (Brankovic et al., 2020).

We are not aware of previous learning theoretic excess risk bounds for clustering problems on curves. Recent learning bounds in the case of clustering points were given in (Bucarelli et al., 2023). An important concept studied in the context of so-called coresets (Phillips, 2017; Munteanu & Schwiegelshohn, 2018; Munteanu, 2023), is the VC dimension of the space of curves under the Fréchet distance (Driemel et al., 2021; Brüning & Driemel, 2023; Cheng & Huang, 2024; Cohen-Addad et al., 2025) and dynamic time warping (Conradi et al., 2024). A series of results has been dedicated to dimensionality reducing random projections for curves (Driemel & Krivošija, 2018; Meintrup et al., 2019; Psarros & Rohde, 2023).

## 2. Definitions

A *curve* in Euclidean space $\mathbb{R}^d$, for $d \in \mathbb{N}$, is a continuous function $\tau : [0, 1] \to \mathbb{R}^d$. A *polygonal curve* is a curve such that there exist a finite number $m \in \mathbb{N}$ of values $0 = t_1 \leq t_2 \leq \ldots \leq t_m = 1$, with $w_i = \tau(t_i)$ which we call *vertices*. Further, for each $i \in \{1, \ldots, m - 1\}$ the segment between the two consecutive vertices $\tau(t_i)$ and $\tau(t_{i+1})$ is an affine line segment, i.e., it holds that

$$\tau(t_i + x) = \left(1 - \frac{x}{t_{i+1} - t_i}\right) \cdot \tau(t_i) + \frac{x}{t_{i+1} - t_i} \cdot \tau(t_{i+1}),$$

for all $x \in [0, t_{i+1} - t_i]$. We can thus fully characterize a curve $\tau$ by defining the sequence of its vertices

$\tau = (w_1, \ldots, w_m)$.[1] We say that such a curve $\tau$ has complexity $m$. For simplicity we write $[m] = \{1, \ldots, m\}$. In this paper, the distance between two points $p, q \in \mathbb{R}^d$ is always their Euclidean distance. Therefore, we write $\|p - q\|$ to denote their distance under the $\ell_2$-norm. We work only with discrete transformations, which are based on the following notion of traversals.

Let $\sigma = (v_1, \ldots, v_{m'})$ and $\tau = (w_1, \ldots, w_{m''})$ be two curves of complexities $m'$ and $m''$ respectively. Then, a traversal $T$ of $\sigma$ and $\tau$ is a sequence that consists of pairs of indices $(i, j)$, which we also call *matchings*, for vertices $v_i \in \sigma$ and $w_j \in \tau$, such that

i) the traversal $T$ starts with $(1, 1)$ and ends with $(m', m'')$, and
ii) the pair $(i, j)$ of $T$ can be followed only by one of $(i + 1, j)$, $(i, j + 1)$ or $(i + 1, j + 1)$.

For simplicity, we write $m = \max\{m', m''\}$. Every traversal is monotonic by construction, cf. item ii). Let $\mathcal{T}$ be the set of all traversals $T$ of $\sigma$ and $\tau$, then the discrete Fréchet distance between $\sigma$ and $\tau$ is defined as

$$d_{dF}(\sigma, \tau) = \min_{T \in \mathcal{T}} \max_{(i,j) \in T} \|v_i - w_j\|,$$

which satisfies all axioms of a metric on the set of curves in Euclidean space when we merge any sequence of equal vertices into a single vertex.[2]

A related dissimilarity measure of two curves is the dynamic time warping (DTW) distance. It considers the sum of distances matched by the traversal instead of the maximum distance. In the literature we find a generalized version, taking the $\ell_q$-norm for $q \in [1, \infty)$ of the vector comprising the Euclidean distances matched by the traversal. We denote it the $\text{DTW}^{(q)}$ distance, which for two curves $\sigma$ and $\tau$ from $\mathbb{R}^d$ is defined as

$$d_{\text{DTW}^{(q)}}(\sigma, \tau) = \min_{T \in \mathcal{T}} \left(\sum_{(i,j) \in T} \|v_i - w_j\|^q\right)^{1/q}.$$

Note that $d_{\text{DTW}^{(q)}}(\sigma, \tau)$ can already be computed if we are only given all pair-wise distances between vertices $(v, w)$ with $v \in \sigma$ and $w \in \tau$. Hence, given any distance matrix $D \in \mathbb{R}^{m' \times m''}$, we define $\text{DTW}^{(q)}(D) = \min_{T \in \mathcal{T}} (\sum_{(i,j) \in T} D[i, j]^q)^{1/q}$. $\text{DTW}^{(1)}$ is most popular across the literature, which is the standard DTW distance. Therefore, we simply use DTW to denote $\text{DTW}^{(1)}$ in the remainder.

$\text{DTW}^{(q)}$ does not satisfy the triangle inequality. However, a relaxed version, depending polynomially on $m$, holds under certain assumptions, as shown by Lemire (2009):

---

[1] We slightly abuse notation using the functional and the sequence representation interchangeably for the same curve $\tau$.

[2] Without this technical assumption, one can define distinct curves with zero distance.

**Lemma 2.1** (Lemma 3 and Theorem 2 of Lemire (2009)). *Given $1 \leq q < \infty$. There exists no constant $c$ that is independent of the complexities of the input curves such that*
$$d_{\mathrm{DTW}^{(q)}}(\sigma, \tau) \leq c \cdot (d_{\mathrm{DTW}^{(q)}}(\sigma, \upsilon) + d_{\mathrm{DTW}^{(q)}}(\upsilon, \tau)).$$
*holds for any three curves $\sigma$, $\tau$, and $\upsilon$ in Euclidean space $\mathbb{R}^d$. If the input curves $\sigma$, $\tau$, and $\upsilon$ are of the same complexity $m$, then it holds that $d_{\mathrm{DTW}^{(q)}}(\sigma, \tau) \leq \sqrt[q]{m} \cdot (d_{\mathrm{DTW}^{(q)}}(\sigma, \upsilon) + d_{\mathrm{DTW}^{(q)}}(\upsilon, \tau)).$*

Inspired by the Ky-Fan norm for matrices, which sums their $k$ largest singular values (Fan, 1951), we propose a novel dissimilarity measure for curves, which we call *k-largest dynamic time warping* distance ($k$-DTW). It seeks for a traversal of the two curves minimizing the sum of the $k$ largest distances between vertices matched by the traversal. We focus on the case $k \ll m$, as we want to observe only a small, yet significant part of the transformation between two curves. If $k$ is larger than the optimal traversal (whose length is at least $m$), then we can simply fill the missing part with zeros. We note that this is just for technical convenience.

**Definition 2.2** ($k$-DTW distance). Given two curves $\sigma = (v_1, \ldots, v_{m'})$ and $\tau = (w_1, \ldots, w_{m''})$ in Euclidean space $\mathbb{R}^d$ and a parameter $k \in \mathbb{N}$, let $\mathcal{T}$ be the set of all traversals $T$ of $\sigma$ and $\tau$. Let the pair $(i, j) \in T$ attain the $l$-th largest distance $s_l^{(T)} = \|v_i - w_j\|$ in $T$ such that $s_1^{(T)} \geq s_2^{(T)} \geq \ldots \geq s_{|T|}^{(T)}$, where $|T|$ denotes the number of pairs in $T$. For any $l > |T|$ let $s_l^{(T)} = 0$. Then, the $k$-DTW distance of $\sigma$ and $\tau$ is defined as
$$d_{k\text{-}DTW}(\sigma, \tau) = \min_{T \in \mathcal{T}} \sum_{l=1}^{k} s_l^{(T)}.$$

The $k$-DTW distance generalizes both, the discrete Fréchet distance (for $k = 1$), and the DTW distance (for $k$ large enough, e.g., $k \geq 2m - 1$).

### 2.1. Triangle inequality

The $k$-DTW distance satisfies a relaxed triangle inequality without assumptions on the curves' complexities, which we prove in the following.

**Lemma 2.3.** *For any three curves $\sigma$, $\tau$, and $\upsilon$ in Euclidean space $\mathbb{R}^d$, it holds that*
$$d_{k\text{-}DTW}(\sigma, \tau) \leq k \cdot (d_{k\text{-}DTW}(\sigma, \upsilon) + d_{k\text{-}DTW}(\upsilon, \tau)).$$

*This bound is tight. Further, there is no constant $c > 0$ independent of $k$ and the complexity of the curves that satisfies $d_{k\text{-}DTW}(\sigma, \tau) \leq c \cdot (d_{k\text{-}DTW}(\sigma, \upsilon) + d_{k\text{-}DTW}(\upsilon, \tau))$ for any three input curves.*

*Proof.* Let $T(\sigma, \upsilon)$ and $T(\upsilon, \tau)$ be traversals that realize $d_{k\text{-}DTW}(\sigma, \upsilon)$ and $d_{k\text{-}DTW}(\upsilon, \tau)$ respectively. Using these traversals, we construct a (not necessarily optimal) traversal

$T'(\sigma, \tau)$ with the underlying idea that any $(i, j) \in T'(\sigma, \tau)$ will correspond to some $(i, l) \in T(\sigma, \upsilon)$ and $(l, j) \in T(\upsilon, \tau)$ such that we can use the triangle inequality. We construct $T'(\sigma, \tau)$ as follows. For any index $z$ of a vertex of $\upsilon$, we consider all vertices that it is matched to in the traversals $T(\sigma, \upsilon)$ and $T(\upsilon, \tau)$, i.e., all $(i_1, z), \ldots, (i_s, z) \in T(\sigma, \upsilon)$ and all $(z, j_1), \ldots, (z, j_t) \in T(\upsilon, \tau)$. Note that by the definition of a traversal, the indices $i_1, \ldots, i_s$ and $j_1, \ldots, j_t$ are non-empty, contiguous subsets of the natural numbers respectively. W.l.o.g., let $s \geq t$; the other case is symmetric. For all $l \in \{1, \ldots, t\}$, we add the tuples $(i_l, j_l)$ to our new traversal $T'(\sigma, \tau)$, and for all $l \in \{t + 1, \ldots, s\}$ we add the tuples $(i_l, j_t)$ to $T'(\sigma, \tau)$. Performing this for all indices $z$ of vertices of $\upsilon$, we obtain a valid traversal $T'(\sigma, \tau)$.

By construction, we know that any element in $T'(\sigma, \tau)$ was created from some elements in $T(\sigma, \upsilon)$ and $T(\upsilon, \tau)$. Furthermore, recall that $s_1^T$ is the largest distance for some given traversal $T$. Hence, by the triangle inequality we obtain $s_1^{T'(\sigma, \tau)} \leq s_1^{T(\sigma, \upsilon)} + s_1^{T(\upsilon, \tau)}$. Using this inequality, we then have
$$
\begin{aligned}
d_{k\text{-}DTW}(\sigma, \tau) &\leq \sum_{l=1}^{k} s_l^{T'(\sigma, \tau)} \leq k \cdot s_1^{T'(\sigma, \tau)} \\
&\leq k \cdot (s_1^{T(\sigma, \upsilon)} + s_1^{T(\upsilon, \tau)}) \\
&\leq k \cdot (d_{k\text{-}DTW}(\sigma, \upsilon) + d_{k\text{-}DTW}(\upsilon, \tau)).
\end{aligned}
$$

At first glance, it might seem that the above inequalities are quite loose. However, we prove in the remainder that they are actually tight in general. To this end, consider the curves $\sigma$, $\tau$, and $\upsilon$, each of complexity $m \geq 3$, similar to the curves that Lemire (2009) introduced for the proof of Lemma 2.1. That is, let $\sigma = (0, \ldots, 0)$, $\tau = (0, \varepsilon, \ldots, \varepsilon, 0)$, and $\upsilon = (0, \varepsilon, 0, \ldots, 0)$. Let $k \leq m - 2$. Then, $d_{k\text{-}DTW}(\sigma, \tau) = k \cdot \varepsilon$, $d_{k\text{-}DTW}(\sigma, \upsilon) = \varepsilon$, and $d_{k\text{-}DTW}(\upsilon, \tau) = 0$, implying that the relaxed triangle inequality that we showed is tight.

Further, towards a contradiction suppose that there is a constant $c > 0$ for which any three curves $\sigma$, $\tau$, and $\upsilon$ satisfy $d_{k\text{-}DTW}(\sigma, \tau) \leq c \cdot (d_{k\text{-}DTW}(\sigma, \upsilon) + d_{k\text{-}DTW}(\upsilon, \tau))$. Then assuming $k \leq m - 2$, the above example implies that $c \geq k$. In the remaining case where $k > m - 2$, the inequality implies $d_{k\text{-}DTW}(\sigma, \tau) = (m - 2) \cdot \varepsilon \leq c \cdot \varepsilon$, thus $c \geq m - 2$. □

### 2.2. Robustness

The concept of robustness for curve distance measures can be quantified as follows to support the intuition: given two curves whose matched vertices are at constant distance, say 1, if we move one vertex away to increase the distance by a large value $\Delta$, then the average distance contributed per vertex increases through this modification by $\Delta / \Theta(k)$ for $k$-DTW, $k \in [m]$. This means that Fréchet is largely dominated by the single outlier, while for $k$-DTW the increase averages out, so that one single large perturbation of order $\Delta \approx \varepsilon k$ is largely indistinguishable from tiny $\varepsilon$

perturbations of (all) single vertices. To make the claim of robustness more rigorous, we follow the outline of (Lopuhaä & Rousseeuw, 1991) and extend it towards a notion of robustness for curves with respect to the $k$-DTW distance (including Fréchet and DTW). To the best of our knowledge, such a notion of robustness for curves has not been considered before.

Given a curve $\pi = (p_1, \ldots, p_m)$ in $\mathbb{R}^d$, let $t_m(\pi) = \sigma$ be its *curve-of-top-$k$-median*, i.e., $\sigma = (s_1, \ldots, s_m)$ where each $s_i$ equals the geometric median restricted to the top-$k$ distances (Krivosija & Munteanu, 2019; Afshani & Schwiegelshohn, 2024) of the set $\{p_1, \ldots, p_m\}$. More formally, for all $j \in [m]$ we define $s_j = \bar{s} \in \arg\min_{s \in \mathbb{R}^d} \sum_{\text{top-}k} \|p_i - s\|$. We note that the $\arg\min$ may be ambiguous. In that case we may choose an arbitrary but fixed element, i.e., we require that $s_1 = \ldots = s_m$. We finally note that $\sum_{\text{top-}k} \|p_i - \bar{s}\| = d_{k\text{-}DTW}(\pi, \sigma)$. It is easy to see that $t_m(\pi)$ is translational equivariant, which means that for any $v \in \mathbb{R}^d$ that we add simultaneously to all vertices of a curve, it holds that $t_m(\pi + v) = t_m(\pi) + v$. We prove this property in Lemma A.9 in the appendix.

First, we define the breakdown point for $t_m(\pi)$ with respect to $k$-DTW to be the smallest number $1 \leq \ell \leq m$ of vertices to obtain $\pi_\ell$, which equals $\pi$ in all but $\ell$ many vertices that may be arbitrarily corrupted, such that $\sigma_\ell = t_m(\pi_\ell)$ is also arbitrarily corrupted. Formally, we define $\beta(t_m, \pi) :=$

$$\min\{\ell \in [m] \mid \sup_{\pi_\ell} d_{k\text{-}DTW}(t_m(\pi), t_m(\pi_\ell)) = \infty\}.$$

We prove the following theorem in Appendix A.3.

**Theorem 2.4.** *Let $\pi = (p_1, \ldots, p_m)$ be a curve with $p_i \in \mathbb{R}^d$. Let $t_m(\pi) = \sigma$ be the curve-of-top-$k$-median. Then $\beta(t_m, \pi) = \left\lfloor \frac{k+1}{2} \right\rfloor$.*

The proof is composed of two parts. Lemma A.10 shows $\beta(t_m, \pi) \leq \left\lfloor \frac{k+1}{2} \right\rfloor$. If $\beta(t_m, \pi)$ were larger than $\left\lfloor \frac{k+1}{2} \right\rfloor$, it implies that any $\pi_\ell$ that we get by corrupting $\ell = \left\lfloor \frac{k+1}{2} \right\rfloor$ vertices would result in bounded $\|t_m(\pi_\ell)\| < \infty$. We construct two such corrupted curves, by translating a suitable choice of $\ell$ vertices by $\pm v$ for a large vector $v$. This requires special care in comparison to (Lopuhaä & Rousseeuw, 1991) to ensure that the set of vertices that determine the top-$k$ distances remains unchanged. The two curves can then be related to each other using the translation equivariance in such a way that the finite bound cannot hold for both simultaneously, thus leading to a contradiction.

Lemma A.11 shows $\beta(t_m, \pi) \geq \left\lfloor \frac{k+1}{2} \right\rfloor$. If that were not true, then we can show that $t_m(\pi_\ell)$ must be close to the original $t_m(\pi)$, as otherwise, it would contradict the optimality of $t_m(\pi_\ell)$ for the corrupted curve. Again, the technical details require special care to account for the geometric median of top-$k$ distances instead of the full set of distances.

## 3. Construction Algorithm

Here we show how to efficiently compute the $k$-DTW distance. We first show in Lemma 3.1 that the $k$-DTW distance between two curves is not equal to taking the largest $k$ distances from the sum that yields their DTW distance. This precludes the option of directly applying the standard DTW algorithm for the computation of $k$-DTW. We also show that the length of the traversals may differ significantly (linearly in $m$) in both directions, which also precludes using a standard DTW algorithm to even estimate the size of a $k$-DTW traversal. This indicates that no provable approximation can be obtained in such a straightforward manner. The proof can be found in Appendix A.

**Lemma 3.1** (Short version of Lemma A.4)**.** *Given an integer $m \geq 5$, it holds for any $k$ with $1 \leq k \leq 4\lfloor \frac{m}{5} \rfloor$, that there exist curves $\sigma$ and $\tau$ of complexity $m$ such that $d_{k\text{-}DTW}(\sigma, \tau)$ is not equal to the sum of the largest $k$ distances contributing to $d_{DTW}(\sigma, \tau)$. Furthermore, the difference between traversal lengths can be linear in the complexity $m$.*

We also stress that the standard dynamic programs for DTW or Fréchet computation cannot be adapted in a direct way to $k$-DTW (i.e., storing the top $k$ distances seen along each path). The problem is that one can update inductively the cost of a current solution in a standard way, but the top $k$ distances cannot be updated to preserve optimality in this way along each path to the end of the dynamic program. Other dynamic programming approaches seem to give only $O(m^8)$ time algorithms (Garfinkel et al., 2006).

As a consequence, we need to design a new algorithm for the $k$-DTW distance. Our solution is given in Algorithm 1 (whose notation we use in the following). It is inspired by Algorithm A of Bertsimas & Sim (2003), designed for the more general framework of top-$k$-optimization. Our analysis is significantly simplified and adapted to the case of the $k$-DTW distance. Our algorithm uses the standard DTW as a subroutine, which can be computed exactly in $O(m'm'')$ time (Berndt & Clifford, 1994; Vintsyuk, 1968).

We give a self contained proof (in Appendix A) in the $k$-DTW context, i.e., we prove that Algorithm 1 returns the correct $d_{k\text{-}DTW}(\sigma, \tau)$, for any two curves $\sigma$ and $\tau$.

**Theorem 3.2.** *Given two curves $\sigma$ and $\tau$, $|\sigma| = m'$ and $|\tau| = m''$, and a parameter $k$, Algorithm 1 returns $d_{k\text{-}DTW}(\sigma, \tau)$ in $O(m'm''z)$ time, where $z$ is the number of distinct distances between any pair of vertices in $\sigma$ and $\tau$.*

Here, we give a high level intuition of the proof. We show that for an arbitrary (not necessarily optimal) but fixed traversal $T$ there exists an iteration where the cost considered by the algorithm equals the $k$-DTW cost of this traversal $T$, i.e., the sum of its $k$ largest distances. In particular this happens in the iteration where the element $E[l^*]$ is a correct guess on the smallest element among the $k$ largest distances in $T$.

---

**Algorithm 1** Computing the $k$-DTW distance

1: **Input:** Curves $\sigma = (v_1 \ldots, v_{m'})$ and $\tau = (w_1, \ldots, w_{m''})$, parameter $k$

2: **Output:** The $k$-DTW distance $d_{k\text{-}DTW}(\sigma, \tau)$

3: Initialize the distance matrix $D[1..m', 1..m'']$ with $D[i,j] \leftarrow \|v_i - w_j\|$

4: Let array $E[1..z+1]$ store the $z$ distinct distances in $D$ including 0 in sorted order $E[1] > \ldots > E[z] > E[z+1] = 0$

5: Initialize mincost $\leftarrow \infty$

6: **for** $l \in \{1, \ldots, z+1\}$ **do**

7:     /* $E[l]$ *represents a current guess on the smallest element among the largest $k$ distances* */

8:     $D'[i,j] \leftarrow \max\{D[i,j] - E[l], 0\}$, for all $(i,j) \in [m'] \times [m'']$

9:     /* *update best solution by the DTW distance on the modified $D'$ matrix plus* $k \cdot E[l]$ */

10:    mincost $\leftarrow \min\{\text{mincost}, \text{DTW}(D') + k \cdot E[l]\}$

11: **end for**

12: **Return:** mincost

---

The reason is that the $\max$ in Line 8 of Algorithm 1 evaluates to $D[i,j] - E[l^*]$ for the $k$ largest distances in $T$ while it evaluates to 0 for all other distances in $T$. Hence, adding $k \cdot E[l^*]$ to the cost of traversal $T$ in $D'$ recovers the original cost. In all other iterations the cost cannot become smaller, which can be shown by the following case distinction:

- For iterations $l$ where $E[l] < E[l^*]$, we again sum the $k$ largest elements of $T$ as, similarly to iteration $l^*$, the $\max$ in Line 8 evaluates to $D[i,j] - E[l]$ for these elements and is compensated for by adding $k \cdot E[l]$. Hence, the cost cannot be less than the $k$-DTW cost of $T$, but it can be larger as we potentially sum more elements.

- For iterations $l$ where $E[l] > E[l^*]$, some of the $k$ largest elements of $T$ are evaluated to 0 in the $\max$ in Line 8. However, by later adding $k \cdot E[l]$, we add at least what we subtracted from $D[i,j]$ in $\max\{D[i,j] - E[l], 0\}$. Hence, again the sum is at least the $k$-DTW cost of $T$.

Applying this to the optimal $k$-DTW traversal $T^*$, we additionally have that the best solution for a suboptimal $T$ cannot ever be smaller than the best and optimal iteration for $T^*$. The minimization in Line 10 thus yields the optimal $k$-DTW cost. The running time follows from the running time of the standard DTW algorithm repeated for $z + 1$ distinct guesses of the element $E[l]$.

Note that $z$ may become as large as $m'm''$ making the running time quartic in bad cases. We thus introduce two heuristic improvements that effectively reduce the number of iterations that need to be performed on practical instances. This reduces the number of DTW calculations on average by $85\% - 97.5\%$ in our experiments in Section 5.

1) The first heuristic leverages the fact that the array of distinct distances $E[1..z+1]$ is sorted and in iteration $l'$ we have a lower bound of $k \cdot E[l']$. Whenever this value exceeds the current mincost value, any element $E[l]$ for $l < l'$ is even larger and cannot yield a better solution. Iterations with $l < l'$ can thus be omitted.

2) For iterations where $E[l]$ is very small, many of the distance entries in $D'$ are larger than zero. This may result in *any* solution being invalid since it is summing over more than $k$ non-zero elements. We can thus run a binary search for the largest index $l'$ that admits at least one valid solution with no more than $k$ non-zero elements and omit all iterations $l > l'$.

Next, we state our rigorous approximation results. The first lemma is a simple, yet important ingredient showing that we can use the discrete Fréchet distance (Eiter & Mannila, 1994) as a $k$-approximation for $k$-DTW. The proof is given in Appendix A.

**Lemma 3.3.** *Given two curves $\sigma = (v_1, \ldots, v_{m'})$ and $\tau = (w_1, \ldots, w_{m''})$, and a parameter $k$, it holds that $d_{dF}(\sigma, \tau)$ is a $k$-approximation for $d_{k\text{-}DTW}(\sigma, \tau)$, which can be computed in time $O(m'm'')$. In particular, it holds that $d_{dF}(\sigma, \tau) \leq d_{k\text{-}DTW}(\sigma, \tau) \leq k \cdot d_{dF}(\sigma, \tau)$.*

Leveraging Lemma 3.3, we obtain our $(1 + \varepsilon)$-approximation result, which quantifies the trade-off between the approximation factor and reducing $z$ to a logarithmic amount, achieving almost quadratic running time.

**Theorem 3.4.** *Given two curves $\sigma$ and $\tau$, $|\sigma| = m'$ and $|\tau| = m''$, and a parameter $k$, there exists a $(1 + \varepsilon)$-approximation algorithm for $k$-DTW for any $0 < \varepsilon \leq 1$ that runs in $O(m'm'' \frac{\log(k/\varepsilon)}{\varepsilon})$ time.*

The proof is given in Appendix A, and we just provide some intuition here. We first compute the discrete Fréchet distance $d_{dF}$. By Lemma 3.3, this $k$-approximation for $k$-DTW allows to round up very small non-zero distances in the distance matrix to $\frac{\varepsilon \cdot d_{dF}}{k}$, which increases the cost by at most $k \cdot \frac{\varepsilon \cdot d_{dF}}{k} \leq \varepsilon \cdot d_{k\text{-}DTW}$. We can also omit iterations where $E[l] > d_{dF}$, because the cost in iteration $l$ is lower bounded by $k \cdot E[l] > k \cdot d_{dF} \geq d_{k\text{-}DTW}$. The remaining distances in $E[1..z]$ are rounded to their next power of $1 + \varepsilon$. This way, no solution becomes cheaper. At the same time, the cost of any solution can increase by at most a factor of $1 + \varepsilon$, and the number of distinct distances is bounded by

$$z \in O(\log_{1+\varepsilon}(k/\varepsilon)) = O(\log(k/\varepsilon)/\varepsilon).$$

We additionally note that the Fréchet distance cannot be approximated within a factor less than 3 in truly subquadratic time unless the Strong Exponential Time Hypothesis (SETH) fails (Buchin et al., 2019c). Hence, computing a good approximation for $k$-DTW has a similar complexity as computing a good approximation for the Fréchet distance.

# 4. Dimension-Free and Improved Learning Theory via $k$-DTW

We now showcase the benefits of using $k$-DTW for learning the median of curves. We are given a distribution $\mathcal{D}$ over polygonal curves of complexity $m$ with vertices in the unit Euclidean ball $B_2^d \subseteq \mathbb{R}^d$. We define $\mathrm{cost}(\mathcal{D}, \psi) := \int_\sigma d(\sigma, \psi)\mathbb{P}[\sigma]\, d\sigma$. Further, let $OPT_\mathcal{D} := \min_\psi \mathrm{cost}(\mathcal{D}, \psi)$. We call the curve $\psi_\mathcal{D}$ inducing the optimal objective $OPT_\mathcal{D}$ the *median curve* with respect to $\mathcal{D}$. Given a sample $P$ of $n$ curves drawn independently and identically distributed from $\mathcal{D}$, we denote by $\psi_P := \mathrm{argmin}_\psi \mathrm{cost}(P, \psi) = \mathrm{argmin}_\psi \sum_{\sigma \in P} d(\sigma, \psi)$ the empirical risk minimizer. Then the generalization error, or the *excess risk* of $\psi_P$ is

$$\mathcal{E} := \mathrm{cost}(\mathcal{D}, \psi_P) - OPT_\mathcal{D}.$$

We are interested in bounding the decrease of $\mathcal{E}$ as a function of the sample size $n$ and problem parameters such as the length of the curves, the ambient dimension, or the new $k$-DTW parameter $k$.

A common way of bounding the generalization error is by means of bounding the Rademacher and Gaussian complexities

$$\mathcal{R}(P) := \mathbb{E} \sup_\psi \left| \frac{1}{n} \sum_{i=1}^n d(\sigma_i, \psi) r_i \right|,$$

$$\mathcal{G}(P) := \mathbb{E} \sup_\psi \left| \frac{1}{n} \sum_{i=1}^n d(\sigma_i, \psi) g_i \right|,$$

where $r_i \in \{-1, 1\}$ are independent Rademacher random variables and $g_i \sim N(0, 1)$ are independent standard Gaussians. It is well-known (Bartlett & Mendelson, 2002) for $P \sim \mathcal{D}$ that there exist absolute constants $c_1 < c_2$ such that

$$\mathbb{E}_P \mathcal{E} \leq c_1 \mathbb{E}_P \mathcal{R}(P) \leq c_2 \mathbb{E}_P \mathcal{G}(P).$$

Thus, it suffices to bound the Gaussian complexity to get the same bound for the Rademacher complexity and the excess risk as well, up to absolute constants.

Before we continue with the main results and their analysis, we would like to point out that the main takeaway message is two-fold: first, to our knowledge, our result gives the first dimension-free bounds for learning clustering of curves, specifically their median, with extensions to $p$-median to be detailed later. Previous bounds for either DTW (Conradi et al., 2024) or Fréchet (Driemel et al., 2021) rely on their respective VC dimension, which is conceptually prone to at least linear dependence on the ambient dimension $d$. Second, while the generalization to the novel $k$-DTW follows in a natural and direct way from our result on DTW, in fact, taking this step together with a lower bound on DTW reveals a non-obvious complexity separation when $k$ is small

enough. More specifically, our results imply that $k$-DTW admits strictly smaller Rademacher and Gaussian complexities than DTW. With this background, we are now ready to state our main learning theoretic results.

**Theorem 4.1.** *Let $\mathcal{D}$ be an arbitrary distribution on curves of complexity $m$ supported over the unit Euclidean ball. Let $P$ be an i.i.d. sample from $\mathcal{D}$ of size $|P| = n$. Then the Rademacher and Gaussian complexities for learning the median curve of complexity $m$ are bounded above by*

$$\mathcal{R}(P), \mathcal{G}(P) \in O\left( \sqrt{\frac{m^3 \cdot \min\{d \log m, m^2 \log^4(mn)\}}{n}} \right)$$

*under DTW and*

$$\mathcal{R}(P), \mathcal{G}(P) \in O\left( \sqrt{\frac{mk^2 \cdot \min\{d \log k, k^2 \log^4(mn)\}}{n}} \right)$$

*under $k$-DTW.*

As a direct consequence, the excess risk of $\psi_P$ is bounded with constant probability by applying Markov's inequality. A high probability bound is possible using a sample complexity of $\tilde{O}(\mathcal{R}(P) + \sqrt{\log(1/\delta)/n})$, see (Bartlett & Mendelson, 2002). We complement this by the following lower bounds on the Rademacher and Gaussian complexity.

**Proposition 4.2.** *Let $\mathcal{D}$ be an arbitrary distribution on curves of complexity $m$ supported over the unit Euclidean ball. Let $P$ be an i.i.d. sample from $\mathcal{D}$ of size $|P| = n$. Then the Rademacher and Gaussian complexities for learning the median curve of complexity $m$ under DTW satisfy $\mathcal{R}(P), \mathcal{G}(P) \in \Omega(\sqrt{m^2/n})$.*

The proof in Appendix A is by reduction from the median problem for points. Comparing the upper and lower bounds, we see that (for sufficiently small $k$) the Rademacher or Gaussian complexity for $k$-DTW is in $\tilde{O}(\sqrt{m/n})$, whereas the corresponding complexity for DTW is in $\Omega(\sqrt{m^2/n})$, showing a *complexity separation* by at least a factor $\tilde{\Omega}(\sqrt{m})$.

We first prove generalization bounds with a dependence on $d$. A full proof of the dimension-free bound is given separately in Appendix A. For a point set $P$, we define $V_{P,DTW}$ to be the set of cost vectors defined by median curves in $B_2^d$ with respect to DTW and we define $V_{P,k\text{-}DTW}$ to be the set of cost vectors defined by median curves in $B_2^d$ with respect to $k$-DTW. That is, for any curve $\psi \subset B_2^d$, there exists a $v^\psi$ with $v_i^\psi = d(\sigma_i, \psi)$, $\sigma_i \in P$, where $d(\cdot, \cdot)$ denotes either DTW or $k$-DTW. Define a net $\mathcal{N}(V_P, \|.\|_\infty, \varepsilon)$ as a set of vectors $N_\varepsilon$ such that for each $\psi \subset B_2^d$, there exists a vector $v' \in \mathcal{N}(V_P, \|.\|_\infty, \varepsilon)$ with $|v_i' - v_i^\psi| \leq \varepsilon$ for all $i \in [|P|]$. We start with the following lemma.

**Lemma 4.3.** *For an absolute constant c, we have that*

*1)* $|\mathcal{N}(V_{P,DTW}, \|.\|_\infty, \varepsilon)| \le \exp(c \cdot d \cdot m \cdot \log(m/\varepsilon))$
*2)* $|\mathcal{N}(V_{P,k\text{-}DTW}, \|.\|_\infty, \varepsilon)| \le \exp(c \cdot d \cdot m \cdot \log(k/\varepsilon))$.

*Proof.* Let $\varepsilon'$ be a constant depending on $\varepsilon$ and $m$, to be determined later. Consider an $\varepsilon'$-net with respect to the $d$-dimensional unit Euclidean ball, that is a covering of $B_2^d$ with Euclidean balls of radius $\varepsilon'$. It is well known that such a covering $C$ using $\left(1 + \frac{2}{\varepsilon'}\right)^d$ balls exists (Pisier, 1999) and can be represented by the set of centers of each ball. We consider all subsets of centers $S \subseteq C$ of size $|S| = m$, inducing an approximate curve $\psi'$. We claim that for an appropriate choice of $\varepsilon'$, the cost vectors induced by the curves $\psi'$ define the desired net. The number of these subsets is upper bounded by $\exp(c \cdot dm \log(1/\varepsilon'))$. Now, we first prove correctness, then reconsider the resulting size. We first observe that for every length $m$ curve $\psi$, there exists a $\psi'$ such that the $j$th vertex of $\psi$ is within distance $\varepsilon'$ of the $j$th vertex of $\psi'$ for each $j \in [m]$, by the properties of $C$. Second, we observe that for any candidate curve $\sigma$ of length at most $m$ and any traversal $T$ of $\sigma$ and $\psi$, we have for all $l \in [|T|]$ that $|s_l^T - s_l^{T\{\psi'\}}| \le \varepsilon'$ by the triangle inequality, where $T\{\psi'\}$ is the traversal $T$ applied to $\sigma$ and $\psi'$ instead of $\psi$. Thus, we have that $\sum_{l=1}^{|T|} |s_l^T - s_l^{T\{\psi'\}}| \le 2m\varepsilon'$. Since this holds for all traversals, it also holds in particular for the optimal traversal. Thus, setting $\varepsilon' = \varepsilon/(2m)$ implies that $|d_{DTW}(\sigma, \psi) - d_{DTW}(\sigma, \psi')| \le \varepsilon$. Rescaling the net accordingly, we obtain $\exp(c \cdot d \cdot m \cdot \log(m/\varepsilon))$.

To extend the argument for $k$-DTW, we merely observe that $\varepsilon'$ only has to be rescaled by a factor $k$, yielding the appropriate bound of $\exp(c \cdot d \cdot m \cdot \log(k/\varepsilon))$. □

The nets provided by Lemma 4.3 already allow us to prove the part of Theorem 4.1 that comes with a dependence on $d$.

*Proof of Theorem 4.1 for low dimensions.* We start by introducing the so-called *chaining* technique. Consider an $n$-dimension vector $v^\psi \in V_P$. Furthermore, let $v^{\psi,j} \in \mathcal{N}(V_P, \|.\|_\infty, 2^{-j})$ be the vector with $\|v^\psi - v^{\psi,j}\|_\infty \le 2^{-j}$ given by Lemma 4.3. Then we may write $v^\psi$ as a telescoping sum with $v^{\psi,0} = 0$

$$v^\psi = \sum_{j=0}^{\infty} v^{\psi,j+1} - v^{\psi,j}.$$

We have due to the triangle inequality $\mathcal{G}(P) :=$

$$\mathbb{E} \sup_\psi \left| \frac{1}{n} \sum_{i=1}^{n} d(\sigma_i, \psi) g_i \right| = \mathbb{E} \sup_\psi \left| \frac{1}{n} \sum_{i=1}^{n} v_i^\psi \cdot g_i \right|$$

$$= \mathbb{E} \sup_\psi \left| \frac{1}{n} \sum_{i=1}^{n} \sum_{j=0}^{\infty} (v_i^{\psi,j+1} - v_i^{\psi,j}) g_i \right|$$

$$\le \sum_{j=0}^{\infty} \mathbb{E} \sup_\psi \left| \frac{1}{n} \sum_{i=1}^{n} (v_i^{\psi,j+1} - v_i^{\psi,j}) g_i \right|.$$

To bound this quantity, observe that $\sum_{i=1}^{n} (v_i^{\psi,j+1} - v_i^{\psi,j}) g_i$ is distributed as a Gaussian $g$ with mean $0$ and its variance $\varsigma_j^2 := \sum_{i=1}^{n} (v_i^{\psi,j+1} - v_i^{\psi,j})^2$ is bounded above by

$$n \cdot \max d(\sigma_i, \psi)^2 \in \begin{cases} O(n \cdot m^2) & \text{for } j = 0, \text{DTW} \\ O(n \cdot k^2) & \text{for } j = 0, k\text{-DTW} \\ O(n \cdot 2^{-2j}) & \text{otherwise.} \end{cases}$$

Thus, using the bounds from Lemma 4.3 and letting $z = m$ for DTW and $z = k$ for $k$-DTW, we have that

$$\sum_{j=0}^{\infty} \mathbb{E} \sup_\psi \left| \frac{1}{n} \sum_{i=1}^{n} (v_i^{\psi,j+1} - v_i^{\psi,j}) g_i \right|$$

$$\le \frac{1}{n} \sum_{j=0}^{\infty} \sqrt{\varsigma_j^2 \log(2|\mathcal{N}(V_P, \|.\|_\infty, 2^{-j})|)}$$

$$\le \frac{1}{n} \sqrt{c \cdot n \cdot d \cdot m \cdot z^2 \log z}$$

$$\quad + \frac{1}{n} \sum_{j=1}^{\infty} \sqrt{c \cdot n \cdot d \cdot m \cdot 2^{-2j} \cdot \log(z 2^j)}$$

$$\le \sqrt{\frac{c' \cdot d \cdot m \cdot z^2 \log z}{n}}$$

where $c, c'$ are absolute constants. Combining both bounds yields the theorem. □

Using a more involved application of the chaining technique, we are able to remove the dependence on $d$, at the cost of an increased dependence on $m$ and $k$. Our proof uses terminal embeddings (Narayanan & Nelson, 2019) as a dimensionality reduction technique. Subsequently, we construct $\varepsilon$-nets akin to Lemma 4.3 within the reduced dimension that is independent of $d$. The convergence argument for the infinite chaining series that worked in the case of low $d$ still applies in a similar, yet adapted way to bound all but the first $j_{\max} = O(\log(n))$ summands. For the remaining terms, we observe that the number of summands is bounded, and each contribution can be bounded again in terms of the net size, which is now independent of $d$. Unfortunately, the vanishing geometric progression $2^{-j}$ cancels with size parameters of the $\varepsilon$-nets, but the additional factor is only $\log(2^{j_{\max}}) = O(\log(n))$, which is affordable in our context.

Finally, we wish to remark that using standard techniques (e.g. the result by Foster & Rakhlin, 2019), these ideas straightforwardly extend to $p$-clustering objectives, generalizing $p$-median to the appropriate problem variant on curves. More specifically, combining our results for the generalization error and Gaussian complexity for median curves with the main result of Foster & Rakhlin (2019) yields a learning rate of $\tilde{O}(\sqrt{p} \cdot \mathcal{G}(P))$. In particular, for the special case of $m = 1$, this recovers the optimal bounds recently proven by Bucarelli et al. (2023) for $p$-median clustering of points, up to polylogarithmic factors.

# 5. Experimental Illustration

In this section, we conduct experiments using our novel $k$-DTW dissimilarity measure to understand the effect of the increased robustness compared to the Fréchet distance and the improved relaxed triangle inequality compared to DTW in practice.[3] To this end, we create a synthetic data set that highlights the properties of the different dissimilarity measures in agglomerative clustering. Subsequently, we perform $l$-nearest neighbor classification on several real-world data sets.

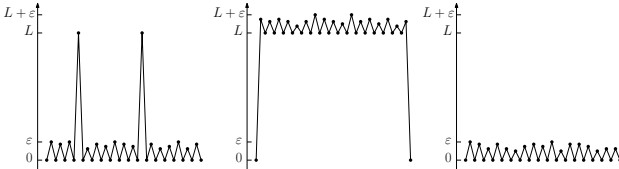

*Figure 1.* Curves of type $A_2$ (left); type $B$ (middle); type $C$ (right)

**Synthetic curves.** Each of the following curves with vertices in $\mathbb{R}$ start and end at 0. We define three types of curves of complexity $4m + 1$, for any $m \in \mathbb{N}$, see Figure 1:

A) Curves of type $A_l$ have $l$ "peaks" (outlier vertices), $l \in \mathbb{N}$. That is, a curve $\alpha \in A_l$ consists of
   - $2m + 1$ vertices with value 0, at indices $2i - 1$ for $i \in [2m + 1]$,
   - $2m - l$ "small value" vertices with value in $[0, \varepsilon]$ at indices $2i$ for $i \in [2m]$, and
   - $l$ peaks of value $L$, at indices $2i$ for $i \in [2m]$.

   The peaks are arbitrarily chosen even indices, cf. Figure 1 (left).

B) Curves of type $B$ have "large values" between $[L, L + \varepsilon]$ at all vertices except for the first and the last, cf. Figure 1 (middle). More precisely, a curve $\beta \in B$ consists of
   - 2 vertices with value 0: $\beta_1$ and $\beta_{4m+1}$,
   - $2m - 1$ vertices with value $L$ at indices $2i + 1$ for all $i \in [2m - 1]$, and
   - $2m$ vertices with value in $[L, L + \varepsilon]$ at indices $2i$, for all $i \in [2m]$.

C) Curves of type $C$ have "small values" for all vertices, cf. Figure 1 (right). More precisely,
   - $2m + 1$ vertices with value 0 at indices $2i - 1$ for all $i \in [2m + 1]$, and
   - $2m$ "small value" vertices with value in $[0, \varepsilon]$ at indices $2i$ for all $i \in [2m]$.

We construct 60 curves, 20 of each type, that demonstrate the advantages of the $k$-DTW distance by choosing appropriate values of $m, \varepsilon, l$, and $L$: the main intuition is that these curves violate the triangle inequality by a $\Theta(m)$ factor for DTW, but only by a $k = O(\log m)$ factor for $k$-DTW.

---

[3]Our Python code is available at https://github.com/akrivosija/kDTW.

Simultaneously, the Fréchet distance suffers from outliers in type-$A_l$ curves for $l = O(1)$, while $k$-DTW is sufficiently robust to compensate spikes by summing over $O(\log m)$ distances.

**Agglomerative Clustering.** Since computing a center based clustering for more than one curve is computationally hard even to approximate in most scenarios – as already mentioned in the introduction – we use a popular alternative called Hierarchical Agglomerative Clustering (HAC), which requires only the pairwise distances between the input data points. The partition process starts with each curve being a singleton cluster. The current clusters are then iteratively merged in ascending order of dissimilarity. Given a dissimilarity measure $d(a, b)$ for two curves, such as Fréchet, DTW, or our novel $k$-DTW, the dissimilarity of two clusters $A, B$ is defined via so called *linkage functions*. The arguably most popular linkage functions are *single linkage* $d(A, B) = \min_{a \in A, b \in B} d(a, b)$ and *complete linkage* $d(A, B) = \max_{a \in A, b \in B} d(a, b)$, where the distance of two clusters is defined as the minimum resp. maximum distance of two input data points taken from the two clusters (Kaufman & Rousseeuw, 1990).

In Figure 2 (top) we see the results for single linkage clustering using the three distance measures. DTW (left) clearly has difficulties to distinguish between type-$A_l$ and type-$C$ curves, since the pairs $(A_l, C)$ are very close, $(A_l, B)$ are moderately close, while $(B, C)$ are far from each other, violating the triangle inequality. The Fréchet distance (right) has very large intra-cluster distances between type-$A_l$ curves that are of the same magnitude as the inter-cluster distances. $k$-DTW (middle) can clearly identify the clusters, as the triangle inequality is less affected, while being robust to the spikes of type-$A_l$ curves. As a result, we obtain pure clusters whose intra-cluster distances are reasonably small and clearly distinguished from the large inter-cluster distances. The results for complete linkage are very similar, cf. Figure 2 (bottom).

*Table 1.* Performance of the Binary Classification of Curves from the Real-World OULAD Data (Kuzilek et al., 2017), for Fréchet, DTW, and $k$-DTW.

| Distance | AUC | Accuracy | $F_1$-score |
|---|---|---|---|
| Fréchet | 0.73737 | 0.83742 | 0.90953 |
| 64-DTW | 0.78795 | **0.85557** | **0.91796** |
| 72-DTW | 0.79639 | 0.85520 | 0.91775 |
| 76-DTW | **0.79772** | 0.85469 | 0.91744 |
| DTW | 0.78360 | 0.85459 | 0.91735 |

**$l$-Nearest Neighbor Classification.** The $l$-nearest neighbor ($l$-NN) model provides a well-known distance based classifier (Devroye et al., 2013). We use real-world data from the Open University Learning Analytics Dataset

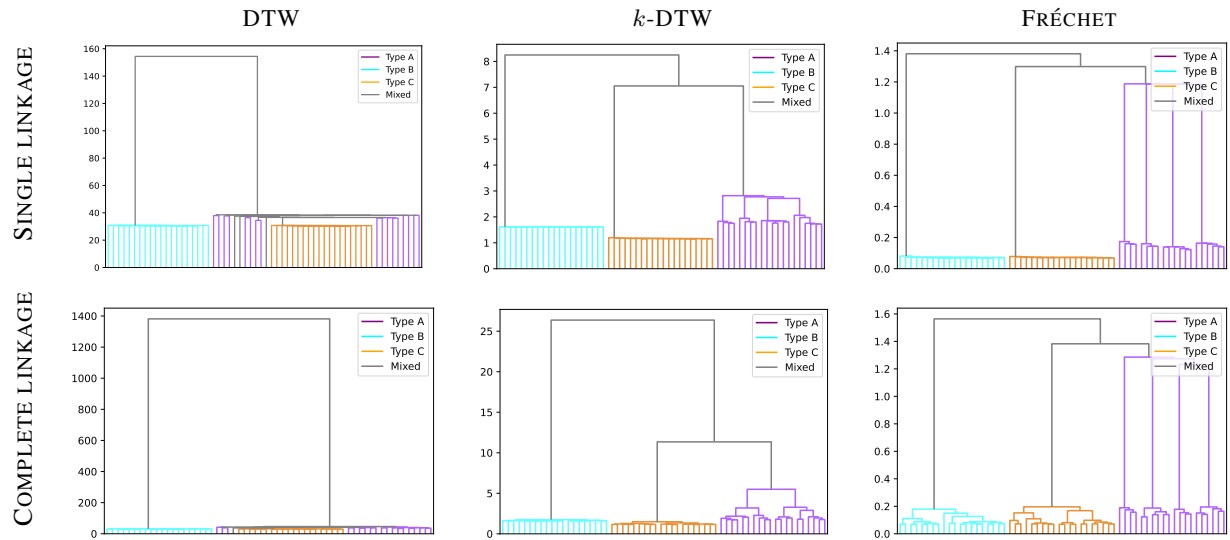

*Figure 2.* Single (top) and complete (bottom) linkage clustering; DTW (left), $k$-DTW (middle), Fréchet distance (right); synthetic data.

(OULAD) (Kuzilek et al., 2017) and the datasets from (Aghababa & Phillips, 2023). OULAD provides click-streams of students acting in an online learning management system. We represent $n = 275$ clickstreams of one course as polygonal curves of complexity $m = 294$. We use $l$-NN in order to predict the final exam result, which serves as a binary label indicating 'pass' or 'fail'. See Appendix B.

We run a 100 times repeated 6-fold cross validation, using $l$-NN with a standard choice of $l = \lceil\sqrt{n}\rceil = 17$ (cf. Devroye et al., 2013). We perform an exponential search for the best parameter $k \in \{2^i \mid i \in [8]\}$ for the $k$-DTW distance, and a subsequent finer search. The best results were obtained for $k \in \{64, 76\}$, which amounts to roughly $20\% - 25\%$ of the curves' complexity. A selection of results is given in Table 1. $k$-DTW outperformed the classification performance of Fréchet and DTW by a margin of up to $8.2\%$ resp. $1.8\%$, when measured by AUC. $k$-DTW also shows slight improvements in terms of accuracy and $F_1$-score. $k = 72$ provides the best values on average over AUC, accuracy, and $F_1$-score. Please refer to Table 2 in Appendix B.

In addition to our explicit parameter search, cross-validated over the full data, we also perform a hold-out evaluation. The parameter $k$ is tuned by cross-validation on training data, and the best $k$ is used on a hold-out test dataset, for $l$-NN classification. Specifically, we randomly split the OULAD dataset (Kuzilek et al., 2017) into training and test sets $A$ and $B$, respectively, where the test set comprises a $\frac{1}{3}, \frac{1}{4}$, or $\frac{1}{5}$ fraction of the input. We find the best value of $k$ on $A$ by running 100 independent repetitions of 6-fold cross-validation, using $l = \lceil\sqrt{|A|}\rceil$ for the $l$-NN training and evaluate it on $B$. Finally, we compare the performances of Fréchet, $k$-DTW and DTW distances on the test set $B$.

The results are presented in Appendix B.3.2.

We finally stress that no extensive search for $k$ is needed. In our experience, it suffices to compare $k$-DTW for few small values of $k \in \{\ln(m), \sqrt{m}, m/10, m/4\}$. At least one $k$-DTW variant yields best or close to the best performance across all classification performance measures. Simultaneously, the worst results are often attained using one of DTW or Fréchet, cf. Tables 7 and 8 in Appendix B. Thus, $k$-DTW yields the best compromise.

Additionally, we perform experiments using several related benchmark distance measures: weak discrete Fréchet distance (Buchin et al., 2019c), edit distance with real penalty (Chen & Ng, 2004), and two variants of partial DTW distance (there are multiple distance measures under the same name in the literature). We use *partial window DTW* (Sakoe & Chiba, 1978), which matches vertices of each curve only within a frame of bounded width $w$, and *partial segment DTW* (Tak & Hwang, 2007; Luo et al., 2024), which partitions both curves into $L$ segments, each of which are matched via standard DTW.

We conclude that $k$-DTW has competitive performance to the gold standards DTW and Fréchet, and further baselines. It is mostly among the best performing distance measures, and sometimes outperforms the competitors. It remains an important open problem to obtain better running times for computing the $k$-DTW distance. In practice, we can speed-up the computation using the DTW heuristic of Silva & Batista (2016). However, solving the challenging problem of improving the top-$k$ optimization framework is an important open question. Its solution would imply provably faster algorithms for $k$-DTW, as well as for other top-$k$ optimization problems.

## Acknowledgements

We thank the anonymous reviewers for their valuable comments. We thank Felix Krall and Tim Novak for their help with implementations and experiments. Amer Krivošija was supported by the project "From Prediction to Agile Interventions in the Social Sciences (FAIR)" funded by the Ministry of Culture and Science MKW.NRW, Germany. Alexander Munteanu was supported by the German Research Foundation (DFG) - grant MU 4662/2-1 (535889065) and by the TU Dortmund - Center for Data Science and Simulation (DoDaS), and acknowledges additional support by the University of Cologne. André Nusser was supported by the French government through the France 2030 investment plan managed by the National Research Agency (ANR), as part of the Initiative of Excellence of Université Côte d'Azur under reference number ANR-15-IDEX-01. Chris Schwiegelshohn was partially supported by the Independent Research Fund Denmark (DFF) under a Sapere Aude Research Leader grant No 1051-00106B.

## Impact Statement

This paper presents work whose goal is to advance the field of Machine Learning. There are many potential societal consequences of our work, none which we feel must be specifically highlighted here.

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

# A. Proofs

## A.1. Algorithmic Part

The discrete Fréchet distance and the $\text{DTW}^{(q)}$ distance between two curves of complexity $m', m''$ can be computed using dynamic programming in $O\left(m'm''\right)$ time (Eiter & Mannila, 1994; Vintsyuk, 1968; Berndt & Clifford, 1994). Here we show how to efficiently compute the $k$-DTW distance.

First, we prove the correctness and the running time of Algorithm 1, that computes the $k$-DTW distance of two given curves.

**Theorem A.1** (Restatement of Theorem 3.2)**.** *Given two curves $\sigma$ and $\tau$, $|\sigma| = m'$ and $|\tau| = m''$, and a parameter $k$, Algorithm 1 returns $d_{k\text{-DTW}}(\sigma, \tau)$ in $O\left(m'm''z\right)$ time, where $z$ is the number of distinct distances between any pair of vertices in $\sigma$ and $\tau$.*

*Proof of Theorem 3.2/A.1.* Consider an arbitrary but fixed traversal $T$. We prove that there exists an iteration $l^*$ where the cost considered by Algorithm 1 equals the exact $k$-DTW cost of $T$, i.e., the sum of the largest $k$ distances in the traversal $T$, which we denote $DTW_k(T)$.

Consider iteration $l = l^*$ where $E[l^*]$ is the smallest element that appears in the sum of largest $k$ elements $DTW_k(T)$. Let $k_l$ be the number of elements of the matrix $D$ that appear in $T$ such that $D[i, j] > E[l]$. Note, that in iteration $l^*$ it holds that $k_{l^*} < k \leq k_{l^*+1}$. We note that the margin case where $k > |T|$ implies, similarly to Definition 2.2, that some distances will be filled with zeros. This in turn implies that $E[l^*] = 0$ and thus $l^* = z + 1$ is the largest possible index that occurs in Algorithm 1. Therefore the upper bound $k_{l^*+1} = k_{z+2}$ is undefined and does not apply in this case.

Further, we have that

$$\text{cost}(T, l^*) := k \cdot E[l^*] + \sum_{(i,j) \in T} \max\{D[i, j] - E[l^*], 0\}$$

$$= (k - k_{l^*}) \cdot E[l^*] + \sum_{\substack{(i,j) \in T, \\ D[i,j] > E[l^*]}} (D[i, j] - E[l^*] + E[l^*]) = \text{DTW}_k(T),$$

since the sum ranges exactly over the $k_{l^*}$ largest elements and no matter how often $E[l^*]$ appears along $T$, it is added exactly $k - k_{l^*}$ times, i.e., the required number of times to complete the sum of $k$ largest elements, which equals $\text{DTW}_k(T)$.

Next, we prove that in all other iterations, the considered $\text{cost}(T, l)$ cannot be smaller than the actual $k$-DTW cost $\text{DTW}_k(T)$. To this end, we make a case distinction below. We define $T_k \subseteq T$ to be an arbitrary but fixed subset indexing $|T_k| = k$ largest elements $D[i, j]$ in $T$, where ties are broken arbitrarily. In the margin case that $k > |T|$, we set $T_k = T$ (the remaining elements are filled with zeros anyway, cf. Definition 2.2). Similarly to the discussion above, if $k > |T|$, no iteration $l > l^*$ exists, and we can proceed directly with the second case where $l < l^*$.

$l > l^*$**:** Note that for all elements $(i, j) \in T_k$ it holds that $D[i, j] > E[l]$, since $k_l \geq k$. Then we have that

$$\text{cost}(T, l) = k \cdot E[l] + \sum_{(i,j) \in T} \max\{D[i, j] - E[l], 0\}$$

$$= k \cdot E[l] + \sum_{(i,j) \in T_k} \max\{D[i, j] - E[l], 0\} + \sum_{(i,j) \notin T_k} \max\{D[i, j] - E[l], 0\}$$

$$= \sum_{(i,j) \in T_k} D[i, j] + \sum_{(i,j) \notin T_k} \max\{D[i, j] - E[l], 0\}$$

$$= DTW_k(T) + \sum_{(i,j) \notin T_k} \max\{D[i, j] - E[l], 0\} \geq DTW_k(T),$$

since the sum over $(i, j) \notin T_k$ is non-negative.

$l < l^*$**:** In this case, we have $k_l < k$ and for all $(i, j) \in T_k \setminus T_{k_l}$ it holds that $E[l] > D[i, j]$. Thus, the sum is taken over less than $k$ elements, and we have that

$$\text{cost}(T, l) = k \cdot E[l] + \sum_{(i,j) \in T} \max\{D[i, j] - E[l], 0\}$$

$$= (k - k_l) \cdot E[l] + \sum_{(i,j) \in T_{k_l}} D[i,j] > \sum_{(i,j) \in T_k} D[i,j] = DTW_k(T).$$

Now, we consider an optimal $k$-DTW traversal $T^*$. By the above arguments the cheapest cost for $T^*$ across all iterations equals the exact $k$-DTW cost $DTW_k(T^*) = OPT$. We also have that the considered cost for any suboptimal $T'$ is never smaller than the actual $k$-DTW cost $DTW_k(T')$. By suboptimality of $T'$, its actual $k$-DTW cost is strictly larger than the optimal $k$-DTW cost of $T^*$. Thus, we have that

$$\min_{l \in [z+1]} \text{cost}(T', l) \geq DTW_k(T') > DTW_k(T^*) = OPT.$$

Consequently, the cost returned by Algorithm 1 equals the optimal $k$-DTW cost of the optimal traversal $T^*$, since it corresponds to the smallest cost considered in the $DTW(D') + k \cdot E[l]$ minimization across all iterations $l \in [z+1]$.

The running time follows since the initialization can be completed by computing all $m'm''$ pairwise distances $D[i,j]$ and inserting them into a binary search tree, skipping duplicates. We thus only keep the $z + 1$ distinct items in $O(m'm'' \log(z))$ time and read them in the sorted order in another $O(z)$ time to build the array $E$. Then, Algorithm 1 runs $z + 1 \leq m'm'' + 1$ iterations and in each iteration it updates all distances in the matrix $D'$, which takes time $O(m'm'')$, and runs one DTW calculation in time $O(m'm'')$. The final cost update takes only constant time. The total time is thus $O(m'm'' \log(z) + z + m'm''z) = O(m'm''z)$. □

We next provide a quadratic time $(1 + \varepsilon)$-approximation. As an ingredient towards this goal, we first show how to obtain a simple $k$-approximation for $k$-DTW by calculating the discrete Fréchet distance in quadratic time.

**Lemma A.2** (Restatement of Lemma 3.3). *Given two curves $\sigma = (v_1, \ldots, v_{m'})$ and $\tau = (w_1, \ldots, w_{m''})$, and a parameter $k$, it holds that $d_{dF}(\sigma, \tau)$ is a $k$-approximation for $d_{k\text{-}DTW}(\sigma, \tau)$, which can be computed in time $O(m'm'')$. In particular, it holds that $d_{dF}(\sigma, \tau) \leq d_{k\text{-}DTW}(\sigma, \tau) \leq k \cdot d_{dF}(\sigma, \tau)$.*

*Proof of Lemma 3.3/A.2.* Let $T_F$ and $T_k$ be traversals of $\sigma$ and $\tau$ realizing their discrete Fréchet and $k$-DTW distances, respectively. Using the notation of Definition 2.2 we have

$$d_{dF}(\sigma, \tau) = s_1^{(T_F)} \leq s_1^{(T_k)} \leq \sum_{l=1}^{k} s_l^{(T_k)} = d_{k\text{-}DTW}(\sigma, \tau) \leq \sum_{l=1}^{k} s_l^{(T_F)} \leq k \cdot s_1^{(T_F)} = k \cdot d_{dF}(\sigma, \tau),$$

The running time of $O(m'm'')$ for computing $d_{dF}(\sigma, \tau)$ was proven in (Eiter & Mannila, 1994). □

Using the $k$-approximation and rounding of distances, we can adapt Algorithm 1 to produce a $(1 + \varepsilon)$-approximation for the $k$-DTW distance.

**Theorem A.3** (Restatement of Theorem 3.4). *Given two curves $\sigma$ and $\tau$, $|\sigma| = m'$ and $|\tau| = m''$, and a parameter $k$, there exists a $(1 + \varepsilon)$-approximation algorithm for any $0 < \varepsilon \leq 1$ that runs in $O(m'm'' \frac{\log(k/\varepsilon)}{\varepsilon})$ time.*

*Proof of Theorem 3.4/A.3.* Let $\varepsilon' = \varepsilon/2$. We perform the following modifications of Algorithm 1:

1) We first compute $d_{dF}(\sigma, \tau)$, which gives a $k$-approximation for $d_{k\text{-}DTW}(\sigma, \tau)$ in time $O(m'm'')$ by Lemma 3.3/A.2. Now, set $d_{\min} = \frac{\varepsilon' \cdot d_{dF}(\sigma, \tau)}{k}$, and $d_{\max} = d_{dF}(\sigma, \tau)$.

   When we build the initial distance matrix $D$, we round all non-zero distances $D[i,j]$ with $0 < D[i,j] < d_{\max}$ to their next higher power of $(1 + \varepsilon')^i \cdot d_{\min}$, for

   $$i \in \left\{ 0, \ldots, \left\lceil \log_{1+\varepsilon'} \left( \frac{d_{\max}}{d_{\min}} \right) \right\rceil \right\} = \left\{ 0, \ldots, \left\lceil \log_{1+\varepsilon'} \left( \frac{k}{\varepsilon'} \right) \right\rceil \right\}.$$

   Through this modification, the cost of each edge and thus of any traversal cannot decrease. However, increasing the costs induces an additive as well as a multiplicative error, which we bound as follows:

i) The sum of up to $k$ distances $0 < D[i,j] \le d_{\min}$ along any traversal $T$ can only increase by at most

$$k \cdot d_{\min} = k \cdot \frac{\varepsilon' \cdot d_{dF}(\sigma, \tau)}{k} = \varepsilon' \cdot d_{dF}(\sigma, \tau),$$

since their next power in the progression is $(1 + \varepsilon')^0 \cdot d_{\min} = d_{\min}$.

ii) Any distance $d_{\min} < D[i,j] < d_{\max}$ can only increase by at most a factor $(1 + \varepsilon')$ by construction. By linearity, this also holds for any sum of these distances.

Using these bounds and Lemma 3.3/A.2, the cost of the optimal traversal after rounding the distances is bounded by

$$(1 + \varepsilon') \cdot d_{k\text{-}DTW}(\sigma, \tau) + \varepsilon' \cdot d_{dF}(\sigma, \tau) \le (1 + 2\varepsilon') \cdot d_{k\text{-}DTW}(\sigma, \tau) = (1 + \varepsilon) \cdot d_{k\text{-}DTW}(\sigma, \tau).$$

2) When we set up the array $E[1..z+1]$, we can omit all indices $l$ where $E[l] > d_{\max} = d_{dF}(\sigma, \tau)$. To see this, recall from the proof of Theorem 3.2/A.1 that the cost of any traversal $T$ in iteration $l$ is lower bounded by $k \cdot E[l]$, since we add this amount to a sum of further non-negative costs. Using this fact together with Lemma 3.3/A.2 again, the cost of $T$ in iteration $l$ is lower bounded by

$$k \cdot E[l] > k \cdot d_{dF}(\sigma, \tau) \ge d_{k\text{-}DTW}(\sigma, \tau),$$

which implies that the cost of $T$ is not optimal in iteration $l$.

All modifications can be performed initially in $O(m'm'')$ time. Further, $E[1..z+1]$ contains only $z \in O(\log_{1+\varepsilon'}(k/\varepsilon')) = O(\frac{\log(k/\varepsilon)}{\varepsilon})$ distinct distances. The running time follows by plugging this into the $O(m'm''z)$ running time bound of Theorem 3.2/A.1. $\square$

Next, we show that the $k$-DTW distance between two curves is not equal to taking the largest $k$ distances from the sum that yields their DTW distance. This precludes the option of simply applying the standard DTW algorithm for the computation of $k$-DTW. We also show that the length of the traversals may differ significantly in both directions, which also precludes using a standard DTW algorithm to estimate the size of a $k$-DTW traversal.

**Lemma A.4** (Full version of Lemma 3.1). *Given a parameter $m \in \mathbb{N}$, it holds that:*

i) *There exist two curves $\sigma$ and $\tau$ of complexity $m$ in Euclidean space $\mathbb{R}^d$, such that the difference $\left|\,|T_{(k-DTW)}| - |T_{(DTW)}|\,\right|$ of the lengths of the optimal $k$-DTW and the optimal DTW traversals of $\sigma$ and $\tau$, denoted $|T_{(k-DTW)}|$ resp. $|T_{(DTW)}|$, is linear in $m$. It can hold both, $|T_{(k-DTW)}| < |T_{(DTW)}|$, or $|T_{(k-DTW)}| > |T_{(DTW)}|$.*

ii) *Assume that $m \ge 5$. Then for any $k$, $1 \le k \le 4\lfloor \frac{m}{5} \rfloor$ there exist curves $\sigma$ and $\tau$ of complexity $m$, s.t. $d_{k\text{-}DTW}(\sigma, \tau)$ is not equal to the sum of the largest $k$ distances contributing to $d_{DTW}(\sigma, \tau)$.*

*Proof of Lemma 3.1/A.4.* We show the first claim of the lemma, that is, that the lengths of the traversals can differ. In the course of the proof, we will also see for the same curves that the value of the $k$-DTW distance need not equal the sum of the $k$ largest distances in a traversal that witnesses the DTW distance. This will provide the arguments to prove the second statement of the lemma.

For the first inequality $|T_{(k-DTW)}| < |T_{(DTW)}|$, let us define $K$-*gadgets* $K(t)$ containing four vertices for each of two 1-dimensional curves $\sigma_{(K(t))}$ and $\tau_{(K(t))}$ as follows, see Figure 3 (left) for an illustration:

$$v_{4t+1} = 4tL + L, \quad v_{4t+2} = 4tL + L - \frac{\varepsilon}{2}, \quad v_{4t+3} = 4tL + L + \frac{\varepsilon}{2}, \quad v_{4t+4} = 4tL + L + \frac{3\varepsilon}{2},$$

$$\text{and} \quad w_{4t+1} = 4tL, \quad w_{4t+2} = 4tL - \frac{\varepsilon}{2}, \quad w_{4t+3} = 4tL + L - \frac{\varepsilon}{2}, \quad w_{4t+4} = 4tL + L + \frac{\varepsilon}{2},$$

where $t \in \mathbb{N} \cup \{0\}$ is the offset, and $L, \varepsilon > 0$, with $\varepsilon = \frac{L}{10}$. Let the two traversals of $\sigma_{(K(t))}$ and $\tau_{(K(t))}$ be (written without the offset $4t$ for simplicity):

$$T_1 \colon (1, 1), (2, 2), (3, 3), (4, 4);$$

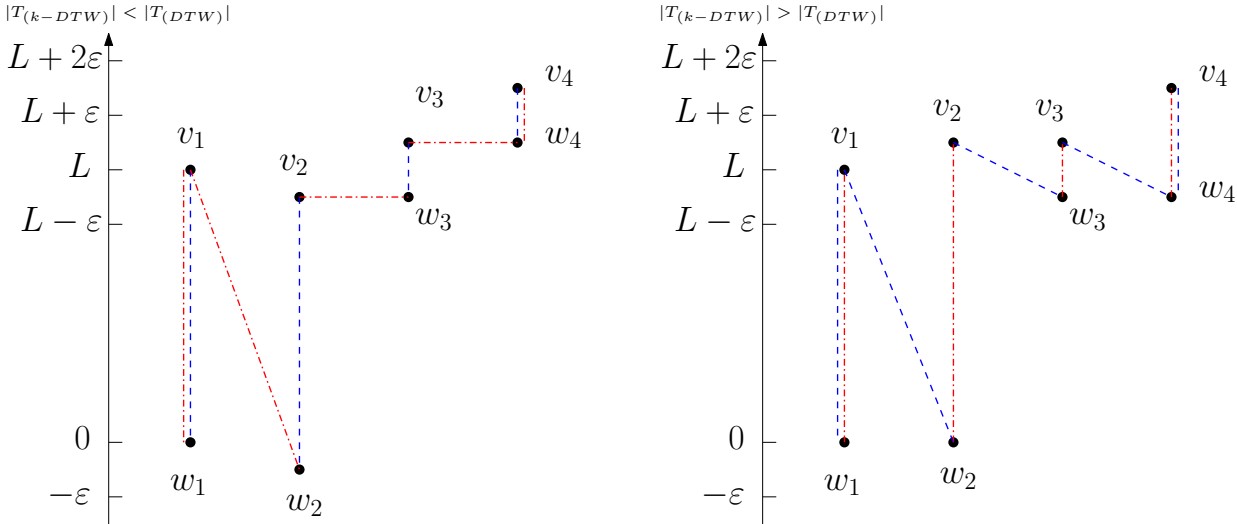

*Figure 3.* $K$-gadget (left); $D$-gadget (right); traversals realizing $k$-DTW (blue); traversals realizing DTW (red).

$$T_2 \colon (1,1), (1,2), (2,3), (3,4), (4,4).$$

It holds that $|T_1| < |T_2| = |T_1| + 1$. The traversal $T_2$ realizes the DTW distance, since

$$\sum_{(i,j) \in T_1} |v_i - w_j| = 2L + 2\varepsilon > 2L + \frac{\varepsilon}{2} + 2 \cdot 0 + \varepsilon = \sum_{(i,j) \in T_2} |v_i - w_j| = d_{DTW}\big(\sigma_{(K(t))}, \tau_{(K(t))}\big).$$

The traversal $T_1$ has the minimum possible traversal size ($|T_1| = 4$), and realizes the $k$-DTW distance (for $k \in \{1, 2, 3\}$ within a single gadget), since for these values of $k$ it holds that $d_{k\text{-}DTW}\big(\sigma_{(K(t))}, \tau_{(K(t))}\big) = \sum_{l=1}^{k} s_l^{(T_1)}$, and the following table shows that $\sum_{l=1}^{k} s_l^{(T_1)} < \sum_{l=1}^{k} s_l^{(T_2)}$ for $k \in \{1, 2, 3\}$.

| $k$ | $\sum_{l=1}^{k} s_l^{(T_1)}$ | $\sum_{l=1}^{k} s_l^{(T_2)}$ |
|---|---|---|
| 1 | $L$ | $L + \frac{\varepsilon}{2}$ |
| 2 | $L + L$ | $(L + \frac{\varepsilon}{2}) + L$ |
| 3 | $L + L + \varepsilon$ | $(L + \frac{\varepsilon}{2}) + L + \varepsilon$ |

We note, that for the value $k = 4$, the inequality $|T_{(k-DTW)}| < |T_{(DTW)}|$ cannot hold (within one gadget) as the $k$-DTW distance cannot be smaller than the sum of all four distances in the traversal. We discuss this further in the proof of the second claim.

Let $\hat{m} = \lfloor \frac{m}{4} \rfloor$, thus $m = 4\hat{m} + r$, where $0 \le r \le 3$. By concatenating the curves $\sigma_{(K(t))}$ and $\tau_{(K(t))}$ from the gadgets $K(0), K(1), \ldots, K(\hat{m} - 1)$ we obtain curves $\sigma$ and $\tau$ of complexity $4\hat{m}$. To get the complexity $m$ we can add $r$ vertices at the end of the both curves, each with value $4\hat{m}L + L$. This does not change the DTW or $k$-DTW distances of $\sigma$ and $\tau$. The traversal obtained by concatenating the traversal $T_2$ repeatedly within each gadget witnesses $d_{DTW}(\sigma, \tau)$, and has length $5\hat{m} + r$.

Note, that edges of the witness traversals cannot belong to two different gadgets, since any two gadgets are almost $3L$ apart. More specifically, a pair $(i, j)$ cannot satisfy $v_i \in K(l)$ and $w_j \in K(l + 1)$ (or $v_i \in K(l + 1)$ and $w_j \in K(l)$) in an optimal traversal for neither DTW nor $k$-DTW. This is because the entire curves $\sigma_{(K(l))}$ and $\tau_{(K(l))}$ are contained in $[4lL - \frac{1}{2}\varepsilon, (4l + 1)L + \frac{3}{2}\varepsilon]$, while $\sigma_{(K(l+1))}$ and $\tau_{(K(l+1))}$ are contained in $[4(l + 1)L - \frac{1}{2}\varepsilon, (4(l + 1) + 1)L + \frac{3}{2}\varepsilon]$. Thus, a pair $(i, j)$ matching vertices across consecutive gadgets would witness a distance of at least $3L - 2\varepsilon > 2L + 2\varepsilon$.

Analogously, the traversal obtained by concatenating the traversal $T_1$ repeatedly in the order of the gadgets, witnesses $d_{k\text{-}DTW}(\sigma, \tau)$ (for $k \in \{1, \ldots, 3\hat{m}\}$), and has length $4\hat{m} + r$. Therefore, the difference of the traversal lengths is

$\left||T_{(k-DTW)}| - |T_{(DTW)}|\right| = \hat{m} = \lfloor \frac{m}{4} \rfloor$, i.e., linear in $m$, which can be made arbitrarily large. In particular, we conclude that there exist instances for which $|T_{(k-DTW)}| < |T_{(DTW)}|$.

To show that there exist instances for which $|T_{(k-DTW)}| > |T_{(DTW)}|$, we define similar $D$-*gadgets* $D(t)$, containing four vertices for each of two curves $\sigma_{(D(t))}$ and $\tau_{(D(t))}$ as follows, see Figure 3 (right) for an illustration:

$$v_{4t+1} = 4tL + L, \quad v_{4t+2} = 4tL + L + \frac{\varepsilon}{2}, \quad v_{4t+3} = 4tL + L + \frac{\varepsilon}{2}, \quad v_{4t+4} = 4tL + L + \frac{3\varepsilon}{2},$$

$$\text{and} \quad w_{4t+1} = 4tL, \quad w_{4t+2} = 4tL, \quad w_{4t+3} = 4tL + L - \frac{\varepsilon}{2}, \quad w_{4t+4} = 4tL + L - \frac{\varepsilon}{2},$$

where $t \in \mathbb{N} \cup \{0\}$ is the offset, and $L, \varepsilon > 0$, with $\varepsilon = \frac{L}{10}$. Let the two traversals of $\sigma_{(D(t))}$ and $\tau_{(D(t))}$ be the same traversals $T_1$ and $T_2$ as in the first claim. Now, the traversal $T_1$ (with the minimum possible traversal size) realizes the DTW distance, since

$$\sum_{(i,j) \in T_2} |v_i - w_j| = 2L + 4\varepsilon > 2L + \frac{\varepsilon}{2} + \varepsilon + 2\varepsilon = \sum_{(i,j) \in T_1} |v_i - w_j| = d_{DTW}\big(\sigma_{(D(t))}, \tau_{(D(t))}\big).$$

The traversal $T_2$ realizes the $k$-DTW distance (for $k \in \{1, 2, 3, 4\}$) within single gadget, since for these values of $k$ it holds that $d_{k\text{-}DTW}\big(\sigma_{(D(t))}, \tau_{(D(t))}\big) = \sum_{l=1}^{k} s_l^{(T_2)}$, and in the following table we see that $\sum_{l=1}^{k} s_l^{(T_1)} > \sum_{l=1}^{k} s_l^{(T_2)}$ holds for $k \in \{1, 2, 3, 4\}$.

| $k$ | $\sum_{l=1}^{k} s_l^{(T_1)}$ | $\sum_{l=1}^{k} s_l^{(T_2)}$ |
|---|---|---|
| 1 | $L + \frac{\varepsilon}{2}$ | $L$ |
| 2 | $(L + \frac{\varepsilon}{2}) + L$ | $L + L$ |
| 3 | $(L + \frac{\varepsilon}{2}) + L + 2\varepsilon$ | $L + L + 2\varepsilon$ |
| 4 | $(L + \frac{\varepsilon}{2}) + L + 2\varepsilon + \varepsilon$ | $L + L + 2\varepsilon + \varepsilon$ |

As in the first part above, vertices in consecutive gadgets cannot be matched since this would cause more than $2L + 4\varepsilon$ cost, which is larger than the whole sum within the gadgets. By concatenating the curves $\sigma_{(D(t))}$ and $\tau_{(D(t))}$ from the gadgets $D(0), D(1), \ldots, D(\hat{m} - 1)$, with additional $r$ vertices $4\hat{m}L + L$ at the end of both $\sigma$ and $\tau$, we obtain curves $\sigma$ and $\tau$ of complexity $4\hat{m} + r$. The traversal obtained by concatenating the traversal $T_1$ repeatedly for all gadgets, witnesses $d_{DTW}(\sigma, \tau)$ and has length $4\hat{m} + r$. Analogously, the traversal obtained by concatenating the traversal $T_2$ repeatedly, witnesses $d_{k\text{-}DTW}(\sigma, \tau)$ (for $k \in \{1, \ldots, 4\hat{m}\}$) and has length $5\hat{m} + r$. Therefore, the difference of the traversal lengths is again $\left||T_{(k-DTW)}| - |T_{(DTW)}|\right| = \hat{m}$, linear in $m$, and can be made arbitrarily large. In particular, we conclude that $|T_{(k-DTW)}| > |T_{(DTW)}|$ in this case. This concludes the proof for the first claim of the lemma.

We next prove the second claim of the lemma, first for $1 \le k < 3\lfloor \frac{m}{4} \rfloor$, and in a second step we extend the upper bound. We discuss the cases $\lfloor \frac{m}{4} \rfloor \le k \le 3\lfloor \frac{m}{4} \rfloor$ and $1 \le k < \lfloor \frac{m}{4} \rfloor$ separately. Again, let $m = 4\hat{m} + r, 0 \le r \le 3$.

$\lfloor \frac{m}{4} \rfloor \le k \le 3\lfloor \frac{m}{4} \rfloor$**:** We can use $\hat{m}$ concatenated K-gadgets from the first part of the lemma to construct the curves $\sigma$ and $\tau$. Both curves are extended at their end by the vertices

$$v_{4\hat{m}+1} = v_{4\hat{m}+2} = v_{4\hat{m}+3} = w_{4\hat{m}+1} = w_{4\hat{m}+2} = w_{4\hat{m}+3} = 4\hat{m}L + L.$$

Hereby, the last (up to) three pairs of vertices $(4\hat{m} + 1, 4\hat{m} + 1)$, $(4\hat{m} + 2, 4\hat{m} + 2)$ and $(4\hat{m} + 3, 4\hat{m} + 3)$ in both traversals $T_1$ and $T_2$ do not interfere with the rest of the matchings for the DTW and the $k$-DTW distance, and do not change the values of $d_{DTW}(\sigma, \tau)$ and $d_{k\text{-}DTW}(\sigma, \tau)$. Then, for all $k$, s.t. $\lfloor \frac{m}{4} \rfloor = \hat{m} \le k \le 3\hat{m} = 3\lfloor \frac{m}{4} \rfloor$, there will be at least one distance among the largest $k$ distances in the $k$-DTW traversal from each gadget. Analogously, there will be at most 3 distances among the largest $k$ in the $k$-DTW traversal. Then, the second claim of the lemma follows from the analysis that we conducted in the first part of the lemma.

$1 \le k < \lfloor \frac{m}{4} \rfloor$**:** To construct the curves $\sigma$ and $\tau$ in this case, we use $k$ concatenated $K$-gadgets from the first part of the lemma, in both curves followed by $m - 4k$ vertices $v_i = w_i = 4kL + L$, for $i \in \{4k + 1, \ldots, m\}$. The traversals $T_1$ and $T_2$ from the first part of the lemma are extended by the matchings $(i, i)$, $i \in \{4k + 1, \ldots, m\}$. Hereby, at least one distance within each of the $k$ gadgets will contribute to the largest $k$ distances in the $k$-DTW traversal. The second claim of the lemma thus follows from the analysis of the first claim.

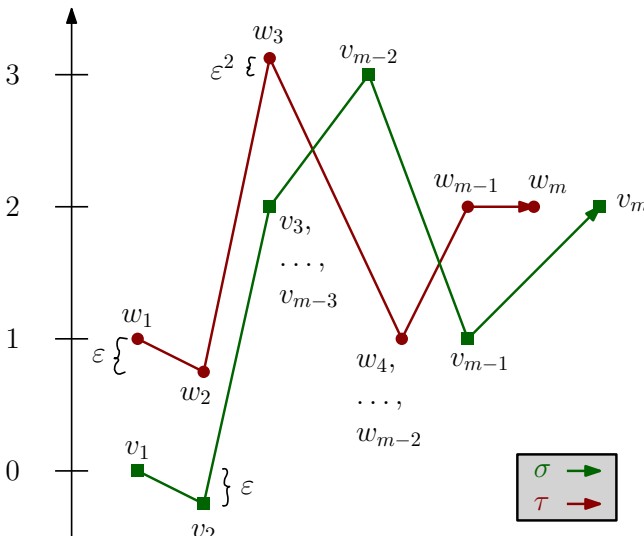

*Figure 4.* Example curves used to prove Lemma A.5. An example instance that has a long DTW traversal (length $2m - 5$) but a short $k$-DTW traversal (length $m + 1$)

The proof of the second claim of the lemma can be repeated analogously using the $D$-gadgets instead of the $K$-gadgets. The following cases need to be considered: $\lfloor \frac{m}{5} \rfloor \le k \le 4\lfloor \frac{m}{5} \rfloor$ and $1 \le k < \lfloor \frac{m}{5} \rfloor$. The rest of the analysis can be conducted verbatim. This completes the proof. $\qquad\square$

We can even show that the traversal lengths of DTW and $k$-DTW can be pushed almost to the extremes $2m$ and $m$, thus maximizing their difference.

**Lemma A.5.** *There exist two curves $\sigma, \tau$ of complexity $m$ with vertices in $\mathbb{R}$ such that the length of the only realizing DTW traversal is $2m - 5$, and there exists a realizing $k$-DTW traversal of length $m + 1$, for any $k \le m - 3$.*

*Proof of Lemma A.5.* We claim that the two curves that realize the properties stated in the lemma are the following, see Figure 4:

$$\sigma = (v_1, \ldots, v_m) = (0, -\varepsilon, \underbrace{2, \ldots, 2}_{m-5}, 3, 1, 2)$$

and

$$\tau = (w_1, \ldots, w_m) = (1, 1 - \varepsilon, 3 + \varepsilon^2, \underbrace{1, \ldots, 1}_{m-5}, 2, 2),$$

where we set $\varepsilon = 1/10$ and choose $m$ large enough, e.g., $m \ge 1000$. Recall that the position $(i, j)$ of a traversal refers to the pair of vertices $(v_i, w_j)$.

To facilitate the understanding, we first give an informal intuition of the DTW and $k$-DTW traversals. Thereafter, we continue by giving a rigorous proof. Due to the small $\varepsilon$-step in the negative direction at the beginning, a $k$-DTW traversal is forced to step forward in both curves since it cannot afford a single matching of cost $1 + \varepsilon$. Consequently, the $k$-DTW traversal needs to continue via the vertices $w_3, \ldots, w_{m-3}$ in order to match the vertices $v_3, \ldots, v_{m-3}$, which can be accomplished in a parallel fashion. The traversal can then be completed using only matchings of cost at most 1. Hence, $k + \varepsilon^2$ is the optimal $k$-DTW cost and it can be realized by an almost parallel traversal of length $m + 1$. For DTW on the other hand, we can leverage that $w_3, \ldots, w_m$ can be matched to $v_{m-2}, v_{m-1}, v_m$ with a cost of only $\varepsilon^2$. Hence, we match $v_1, \ldots, v_{m-3}$ to $w_1, w_2$ with a cost of $m - 3 + 2\varepsilon$ and then transition into the position $(m - 2, 3)$ to match the remainder with a cost of $\varepsilon^2$. If we step forward earlier on $\tau$, then either the matching to $v_3, \ldots, v_{m-3}$ or the matching of vertices $w_{m-2}, w_{m-1}, w_m$ and $v_{m-2}, v_{m-1}, v_m$ at the end becomes too expensive.

We show the claims for DTW and $k$-DTW separately, starting with the latter.

$k$**-DTW:**  Let $k \leq m - 3$. We first show an upper bound of $k + \varepsilon^2$ on the $k$-DTW cost, which is simply given by the following traversal of $\sigma$ and $\tau$:

$$(1, 1), \ldots, (m - 3, m - 3), (m - 3, m - 2), (m - 2, m - 1), (m - 1, m), (m, m).$$

The only matching cost larger than 1 is induced by matching $v_3$ and $w_3$ with a cost of $1 + \varepsilon^2$. Note that this traversal has length $m + 1$.

To show that this upper bound is tight, we first show the following claim: *Consider any traversal and the first time that it reaches $v_{m-3}$. The sum of the $k$ largest matchings of any such partial traversal is at least $k + \varepsilon^2$.*

First, note that the only $w \in \tau$ that have distance $|w - v| < 1$ for any $v \in \{v_1, \ldots, v_{m-3}\}$ are $w_{m-1}, w_m$, and $w_2$ for which only $|v_1 - w_2| < 1$. Note that $v_1$ and $w_2$ are only matched in a traversal that starts with $(1, 1), (1, 2)$, but then any traversal of sufficiently low cost needs to continue with $(2, 2)$. Hence, going directly from $(1, 1)$ to $(2, 2)$ is always better. The only way to continue the traversal from here is to continue the parallel traversal, as otherwise we would incur a matching cost of more than $1 + \varepsilon^2$ or multiple costs of $1 + \varepsilon^2$, which both are prohibitive for a $k$-DTW cost of $k + \varepsilon^2$. It follows from the above observations that any matching to $v_1, \ldots, v_{m-3}$ will lead to a $k$-DTW cost that is at least $k + \varepsilon^2$. This establishes a lower bound of $k + \varepsilon^2$ on the $k$-DTW cost, which matches the upper bound.

**DTW:**  We first show that the DTW cost is at most $m - 3 + 2\varepsilon + \varepsilon^2$ and this is realized by a traversal of length $2m - 5$. The following traversal of $\sigma$ and $\tau$ has these properties:

$$(1, 1), \ldots, (m - 4, 1), (m - 3, 2), (m - 2, 3), (m - 1, 4), \ldots, (m - 1, m - 2), (m, m - 1), (m, m). \tag{1}$$

We now proceed with proving that the DTW cost is also at least $m - 3 + 2\varepsilon + \varepsilon^2$ and is only realized by the above or a longer traversal. Consider which prefixes of $\tau$ can be matched to $v_1, \ldots, v_{m-3}$ with a cost that does not exceed $C = m - 3 + 2\varepsilon + \varepsilon^2$. To this end, first note that completing a traversal from any position $(m - 3, l)$, with $l \geq 4$, has cost at least 2, while any matching of the prefixes $v_1, \ldots, v_{m-3}$ and $w_1, \ldots, w_l$ has at least $m - 3$ pairs at distance at least 1. Thus, the total cost would be at least $m - 2 > C$ for our choice of $m$ and $\varepsilon$. Hence, we have to match one of the vertices $w_1, w_2, w_3$ to $v_{m-3}$. As $w_2$ and $w_3$ have distance $1 + \varepsilon$, resp. $1 + \varepsilon^2$, to $v_3, \ldots, v_{m-3}$, matching them to any subsequence of $v_3, \ldots, v_{m-3}$ of size larger than $1/3\varepsilon$ leads to a total matching cost larger than $C$. Thus, any possible traversal that has cost at most $C$ necessarily matches the majority of $v_3, \ldots, v_{m-3}$ to $w_1$. If the position $(m - 3, 1)$ would be part of such a traversal, then all pairs $(i, 1)$, $i \in [m - 3]$, and the pair $(m - 3, 2)$ would have to be in the traversal as well, which again has cost more than $C$. Therefore, the position $(m - 3, 2)$ must belong to the traversal, causing the cost $1 + \varepsilon$. But adding even one additional position $(l, 2)$, where $l \in [3, m - 4]$, into the traversal causes total cost larger than $C$. Hence, the only remaining traversal with a cost of at most $C = m - 3 + 2\varepsilon + \varepsilon^2$ is the one that realizes the upper bound given in Equation (1). $\qquad \square$

### A.2. Learning Theory

Using a more involved application of the chaining technique, we are able to reduce the dependence on $d$, at the cost of an increased dependence on $m$ and $k$. To this end, we require terminal embeddings defined as follows.

**Definition A.6** (Terminal Embeddings). For a point set $P \subset \mathbb{R}^d$, an $\varepsilon$-terminal embedding is a mapping $f : \mathbb{R}^d \to \mathbb{R}^h$ such that for all $p \in P$ and all $q \in \mathbb{R}^d$

$$(1 - \varepsilon) \cdot \|p - q\| \leq \|f(p) - f(q)\| \leq (1 + \varepsilon)\|p - q\|.$$

Terminal embeddings are similar to Johnson-Lindenstrauss type embeddings, albeit stronger in the sense that $q$ can be any point in $\mathbb{R}^d$, rather than just applying to pairwise distances. Nevertheless, the target dimensions of Johnson-Lindenstrauss embeddings and terminal embeddings are essentially identical, as proven by Narayanan & Nelson (2019) and summarized in the following lemma.

**Lemma A.7.** *There exist terminal embeddings of target dimension* $h \in O(\varepsilon^{-2} \log |P|)$.

An immediate consequence of the terminal embedding guarantee is that both DTW and $k$-DTW are preserved up to $(1 \pm \varepsilon)$ factors, if the target dimension is in $O(\varepsilon^{-2} \log(mn))$. In the following, we show that they also yield an alternative bound on the net size for median curves.

**Lemma A.8.** *For an absolute constant c, we have that*

$$|\mathcal{N}(V_{P,DTW}, \|.\|_\infty, \varepsilon)| \leq \exp(c \cdot m^3 \log^2(mn/\varepsilon) \cdot \varepsilon^{-2})$$

*and*

$$|\mathcal{N}(V_{P,k\text{-}DTW}, \|.\|_\infty, \varepsilon)| \leq \exp(c \cdot m \cdot k^2 \cdot \log^2(mnk/\varepsilon) \cdot \varepsilon^{-2}).$$

*Proof.* Let $z = m$ for DTW and $z = k$ for $k$-DTW. We first apply a terminal embedding of target dimension $h \in O(z^2 \cdot \varepsilon^{-2} \log(mn))$, which preserves DTW and $k$-DTW up to $(1 \pm \varepsilon/z)$ factors, and hence both up to an additive error of $\varepsilon$ as all the points on the curves are assumed to have at most unit Euclidean norm. Subsequently, we apply Lemma 4.3, yielding $2\varepsilon$-nets of size

$$|\mathcal{N}(V_{P,DTW}, \|.\|_\infty, \varepsilon)| \leq \exp(c \cdot h \cdot m \cdot \log \frac{m}{\varepsilon})$$

and

$$|\mathcal{N}(V_{P,k\text{-}DTW}, \|.\|_\infty, \varepsilon)| \leq \exp(c \cdot h \cdot m \cdot \log \frac{k}{\varepsilon}).$$

Inserting the appropriate bounds for $h$ and rescaling $\varepsilon$ yields the claim. $\square$

With this lemma, we now give the dimension-free bound on the Rademacher and Gaussian complexity claimed in Theorem 4.1.

*Proof of Theorem 4.1 for high dimensions.* As with the proof of the low-dimensional case, we rely on the chaining technique. There is, however, a key difference. Let $j_{\max}$ be chosen such that $2^{-j_{\max}} = 1/\sqrt{n}$. We then have due to the triangle inequality

$$\mathcal{G}(P) := \mathbb{E} \sup_\psi \left| \frac{1}{n} \sum_{i=1}^n d(\sigma_i, \psi) g_i \right| \leq \sum_{j=0}^{j_{\max}} \mathbb{E} \sup_\psi \left| \frac{1}{n} \sum_{i=1}^n (v_i^{\psi,j+1} - v_i^{\psi,j}) g_i \right| \tag{2}$$

$$+ \sum_{j=j_{\max}}^{\infty} \mathbb{E} \sup_\psi \left| \frac{1}{n} \sum_{i=1}^n (v_i^{\psi,j+1} - v_i^{\psi,j}) g_i \right|. \tag{3}$$

We bound Equation (2) and Equation (3) separately. For the latter, i.e., Equation (3), observe that

$$\sum_{i=1}^n (v_i^{\psi,j+1} - v_i^{\psi,j})^2 \leq n \cdot 2^{-2j}.$$

Applying the Cauchy-Schwarz inequality to Equation (3) and inserting this bound then yields

$$\sum_{j=j_{\max}}^{\infty} \mathbb{E} \sup_{\psi} \left| \frac{1}{n} \sum_{i=1}^{n} (v_i^{\psi,j+1} - v_i^{\psi,j}) g_i \right| \le \sum_{j=j_{\max}}^{\infty} \frac{1}{n} \mathbb{E} \sqrt{\sum_{i=1}^{n} g_i^2} \cdot \sqrt{\sum_{i=1}^{n} (v_i^{\psi,j+1} - v_i^{\psi,j})^2}$$

$$\le \sum_{j=j_{\max}}^{\infty} 2^{-j} \le 2 \cdot 2^{-j_{\max}} \in O(1/\sqrt{n}).$$

For the former, i.e., Equation (2), we observe that the variance bounds derived in the proof of the low dimensional case are still valid. That is, we have that

$$\varsigma_j^2 := \sum_{i=1}^{n} (v_i^{\psi,j+1} - v_i^{\psi,j})^2 \le n \cdot \max d(\sigma_i, \psi)^2 \in \begin{cases} O(n \cdot m^2) & \text{for } j = 0, \text{ and DTW} \\ O(n \cdot k^2) & \text{for } j = 0, \text{and } k\text{-DTW} \\ O(n \cdot 2^{-2j}) & \text{otherwise.} \end{cases}$$

Thus for some absolute constant $c$, for every $j > 0$

$$\mathbb{E} \sup_{\psi} \left| \frac{1}{n} \sum_{i=1}^{n} (v_i^{\psi,j+1} - v_i^{\psi,j}) g_i \right| \le \frac{1}{n} \sqrt{\varsigma_j^2 \log(2|\mathcal{N}(V_P, \|.\|_\infty, 2^{-j})|)}$$

$$\le \sqrt{\frac{c \cdot m \cdot z^2 \cdot 2^{-2j} \cdot \log^2(mnz2^j) \cdot 2^{2j}}{n}}.$$

and for $j = 0$

$$\mathbb{E} \sup_{\psi} \left| \frac{1}{n} \sum_{i=1}^{n} (v_i^{\psi,1} - v_i^{\psi,0}) g_i \right| \le \sqrt{\frac{c \cdot m \cdot z^4 \cdot \log^2(mnz)}{n}}.$$

Thus, we have

$$\sum_{j=0}^{j_{\max}} \mathbb{E} \sup_{\psi} \left| \frac{1}{n} \sum_{i=1}^{n} (v_i^{\psi,j+1} - v_i^{\psi,j}) g_i \right| \le (j_{\max} + 1) \cdot \sqrt{\frac{c \cdot m \cdot z^4 \cdot \log^2(mnz2^{j_{\max}})}{n}}$$

$$\in O\left( \sqrt{\frac{c \cdot m \cdot z^4 \cdot \log^4(mn)}{n}} \right),$$

since $z \in O(m)$. Combining both bounds then yields the claim. $\qquad \square$

*Proof of Proposition 4.2.* We first recall the corresponding bounds for median queries, that is, for curves of complexity 1. By tightness of Chernoff bounds (see for example Lemma 4 of Klein & Young (2015)), these queries have a Rademacher and Gaussian complexity of $\Omega(\sqrt{1/n})$. Now for every such query, consider a function space for which every point is duplicated $m$ times, that is the point (or complexity 1 curve) $\sigma$ becomes a complexity $m$ curve $\sigma_m$. In this case $d_{DTW}(\sigma_m, \psi_m) = m \cdot \|\sigma - \psi\|$. Thus, every cost vector is scaled by exactly a factor $m$. By linearity of Rademacher and Gaussian complexity, this likewise increases these complexities by a factor of $m$, yielding the claimed bound of $\Omega(\sqrt{m^2/n})$. $\qquad \square$

### A.3. Robustness

We first observe that the sum of top-$k$ elements of a (non-negative) vector $v \in \mathbb{R}^d$ is indeed a norm. This is immediate from the Ky-Fan norm of diagonal matrices $\mathrm{diag}(v) \in \mathbb{R}^{d \times d}$ whose diagonal entries are the coordinates of $v$. This holds since the Ky-Fan norm of order $k$ is the sum of $k$ largest singular values, which correspond to the $k$ largest values in $v$. This correspondence has been exploited in recent loosely related work on sketching and $k$-sparsity in machine learning (Clarkson & Woodruff, 2015; Munteanu et al., 2021; 2022; 2023; Mai et al., 2023; Munteanu & Omlor, 2024) that inspired $k$-DTW.

We follow the outline of (Lopuhaä & Rousseeuw, 1991, Section 2) and extend it towards a notion of the robustness for curves with respect to the $k$-DTW distance (along with its special cases, Fréchet and DTW). To the best of our knowledge, such a notion of robustness for curves has not been introduced or analyzed before.

Given a curve $\pi = (p_1, \ldots, p_m)$ in $\mathbb{R}^d$, let $t_m(\pi) = \sigma$ be its *curve-of-top-$k$-median*, i.e., $\sigma = (s_1, \ldots, s_m)$ where each $s_i$ equals the geometric median restricted to the top-$k$ distances (Krivosija & Munteanu, 2019; Afshani & Schwiegelshohn, 2024) of the set $\{p_1, \ldots, p_m\}$. More formally, for all $j \in [m]$ we define $s_j = \bar{s} \in \mathrm{argmin}_{s \in \mathbb{R}^d} \sum_{\text{top-}k} \|p_i - s\|$. We note that the $\mathrm{argmin}$ may be ambiguous. In that case we may choose an arbitrary but fixed element, i.e., we require that $s_1 = \ldots = s_m$. We finally note that $\sum_{\text{top-}k} \|p_i - \bar{s}\| = d_{k\text{-}DTW}(\pi, \sigma)$.

It is easy to see that $t_m(\pi)$ is translational equivariant, which means that for any $v \in \mathbb{R}^d$ that we add simultaneously to all vertices of a curve, it holds that $t_m(\pi + v) = t_m(\pi) + v$. We prove this property in Lemma A.9 below.

First, we define the breakdown point for $t_m(\pi)$ with respect to $k$-DTW to be the smallest number $1 \leq \ell \leq m$ of vertices to obtain $\pi_\ell$ that equals $\pi$ in all but $\ell$ many vertices that may be arbitrarily corrupted, such that $\sigma_\ell = t_m(\pi_\ell)$ is also arbitrarily corrupted. More formally, we define

$$\beta(t_m, \pi) = \min\{\ell \in [m] \mid \sup_{\pi_\ell} d_{k\text{-}DTW}(t_m(\pi), t_m(\pi_\ell)) = \infty\}. \tag{4}$$

Obviously, it holds that $d_{k\text{-}DTW}(t_m(\pi), t_m(\pi_\ell)) = d_{k\text{-}DTW}(\sigma, \sigma_\ell) = k \cdot \|s_1 - (\sigma_\ell)_1\|$. Since $k$ is always finite for finite curves, the supremum is infinite if and only if the distance between the top-$k$-medians of corrupted and uncorrupted curves is infinite.

**Lemma A.9.** $t_m(\pi)$ *is translational equivariant and as a consequence $\beta(t_m, \pi)$ is also translational equivariant.*

*Proof.* Since $t_m$ is determined by some choice of $k$ vertices out of the $m$ vertices of $\pi$, we can simply restrict to one arbitrary but fixed choice of top-$k$ indices $S \subseteq [m]$ that determines the minimum. Then for any $s \in \mathbb{R}^d$ and any translation $v \in \mathbb{R}^d$, we have that $\sum_{i \in S} \|p_i - s\| = \sum_{i \in S} \|(p_i + v) - (s + v)\|$. In particular this also holds for the minimizer. It thus follows that $t_m(\pi + v) = t_m(\pi) + v$.

For the second claim, note that $\|t_m(\pi + v) - t_m(\pi_\ell + v)\| = \|t_m(\pi) + v - (t_m(\pi_\ell) + v)\| = \|t_m(\pi) - t_m(\pi_\ell)\|$. This means that the supremum over all $\pi_\ell$ that differ from $\pi$ in $\ell$ vertices is infinite if and only if the supremum over all $\pi'_\ell = \pi_\ell + v$ that differ from $\pi' = \pi + v$ in $\ell$ vertices is infinite. $\square$

Our aim is to prove that $\beta(t_m, \pi) = \lfloor \frac{k+1}{2} \rfloor$. We start with the upper bound.

**Lemma A.10.** *Let $\pi = (p_1, \ldots, p_m)$ be a curve with $p_i \in \mathbb{R}^d$. Let $t_m(\pi) = \sigma$ be the curve-of-top-$k$-median. Then $\beta(t_m, \pi) \leq \lfloor \frac{k+1}{2} \rfloor$.*

*Proof.* Since $t_m(\pi)$ is translational equivariant by Lemma A.9, we can assume w.l.o.g. that $t_m(\pi) = (0, \ldots, 0)$ of length $m$. For the sake of a contradiction, suppose that $\beta(t_m, \pi) > \lfloor \frac{k+1}{2} \rfloor$. This implies that any $t_m(\pi_\ell)$, where $\pi_\ell$ is obtained from $\pi$ by corrupting $\ell = \lfloor \frac{k+1}{2} \rfloor$ many vertices arbitrarily, must be bounded. That is, there exists $L \in \mathbb{R}$ such that

$$\|t_m(\pi_\ell)\|_{\text{top-}k} \leq L < \infty.$$

Consider a vector $v \in \mathbb{R}^d$, and let $E_+$ be the set of $\ell$ vertices of $\pi$ that have largest $\langle p_i, v \rangle$ and similarly let $E_-$ be the set of $\ell$ vertices of $\pi$ that have largest $\langle p_i, -v \rangle$ in the opposite direction. These are the points that are furthest away from zero in both directions along the line spanned by $v$. Ties are broken according to the largest distance to the line spanned by $v$ and arbitrarily if ties persist. Let $E = E_+ \cup E_-$.

Now, consider the curve $\pi_{(+v)}$, that we obtain from $\pi$ by adding $v$ to the $\ell$ vertices in $E_+$. Similarly consider the curve $\pi_{(-v)}$, that we obtain from $\pi$ by subtracting $v$ from the $\ell$ vertices in $E_-$. The other vertices remain untouched.

If we let $\|v\|$ grow large enough, then the top-$k$ summands that determine either of $t_m(\pi_{(+v)}), t_m(\pi_{(-v)})$ are dominated by the $2\ell \geq k$ extreme elements in $E$ and the other vertices $p_i \notin E$ do not have an influence. We can thus simply remove the vertices $p_i \notin E$. Denote by $\bar{\pi}_{(+v)}, \bar{\pi}_{(-v)}$ the truncated sequences.

Since the number of corruptions are $\ell$ in each of the constructed curves, it follows from our assumption that $\|t_m(\pi_{(+v)})\|_{\text{top-}k} = \|t_m(\bar{\pi}_{(+v)})\|_{\text{top-}k} \leq L$ and $\|t_m(\pi_{(-v)})\|_{\text{top-}k} = \|t_m(\bar{\pi}_{(-v)})\|_{\text{top-}k} = \|t_m(\bar{\pi}_{(+v)}) - v\|_{\text{top-}k} \leq L$ holds for some $L < \infty$. Putting both bounds together, we get that

$$2L \geq \|t_m(\bar{\pi}_{(+v)}) - v\|_{\text{top-}k} + \|t_m(\bar{\pi}_{(+v)})\|_{\text{top-}k} \geq k \cdot \|v\| - \|t_m(\bar{\pi}_{(+v)})\|_{\text{top-}k} + \|t_m(\bar{\pi}_{(+v)})\|_{\text{top-}k} = k \cdot \|v\|.$$

Thus, any large enough corruption vector that satisfies $\|v\| > \frac{2L}{k}$ yields a contradiction. $\square$

It remains to prove the lower bound.

**Lemma A.11.** *Let $\pi = (p_1, \ldots, p_m)$ be a curve with $p_i \in \mathbb{R}^d$. Let $t_m(\pi) = \sigma$ be the curve-of-top-$k$-median. Then $\beta(t_m, \pi) \geq \lfloor \frac{k+1}{2} \rfloor$.*

*Proof.* Since $t_m(\pi)$ is translational equivariant by Lemma A.9, we can assume w.l.o.g. that $t_m(\pi) = (0, \ldots, 0)$ of length $m$. Let $M = \max_{i \in [m]} \|p_i\|$, and let $B(0, 2M)$ be the ball of radius $2M$ centered at $0$. Let $\pi_\ell := (q_1, \ldots, q_m)$ be a corrupted curve, obtained from $\pi$ by replacing at most $\ell = \lfloor \frac{k-1}{2} \rfloor$ of its vertices by arbitrary vectors. Let $t_m(\pi_\ell)$ be a curve $\hat{\sigma} = (\hat{s}, \ldots, \hat{s})$, where $\hat{s}$ minimizes $\sum_{\text{top-}k} \|q_i - \hat{s}\|$, and let $S \subseteq [m]$ be a set of $k$ indices that determine the minimum.

We show that $\sup_{\hat{\sigma}} \|t_m(\pi_\ell)\|_{\text{top-}k}$, taken over all possible $\hat{\sigma}$, is finite. We may assume that $\hat{s} \notin B(0, 2M)$ since otherwise $\|t_m(\pi_\ell)\|_{\text{top-}k} \leq 2Mk$ is trivially finite. Let $D = \inf_{v \in B(0, 2M)} \|\hat{s} - v\|$ be the smallest distance between $\hat{s}$ and $B(0, 2M)$. Then it holds that

$$\frac{\|t_m(\pi_\ell)\|_{\text{top-}k}}{k} = \|\hat{s}\| \leq (D + 2M).$$

For any vertex $q_i$ in $\pi_\ell$ that is corrupted, it holds by triangle inequality that

$$\|q_i - \hat{s}\| \geq \|q_i\| - \|\hat{s}\| \geq \|q_i\| - (D + 2M). \tag{5}$$

Let us assume that $D$ is large, that is, $D > 2M \cdot \lfloor \frac{k-1}{2} \rfloor$. Since $\pi_\ell \in B(0, M)$, for each of the uncorrupted vertices $q_i = p_i$ of $\pi_\ell$ (there are at least $m - \ell$ of them), it holds that:

$$\|p_i - \hat{s}\| \geq D + M \geq D + \|p_i\|. \tag{6}$$

The top-$k$ values are attained by at most $\ell = \lfloor \frac{k-1}{2} \rfloor$ corrupted vertices, and at least $k - \ell = k - \lfloor \frac{k-1}{2} \rfloor$ uncorrupted ones. Thus, from Equations (5) and (6) it holds that

$$\begin{aligned}
\sum_{\text{top-}k} \|q_i - \hat{s}\| &= \sum_{i \in S} \|q_i - \hat{s}\| \\
&= \sum_{\substack{i \in S, \text{ uncorrupted}}} \|p_i - \hat{s}\| + \sum_{\substack{i \in S, \text{ corrupted}}} \|q_i - \hat{s}\| \\
&\geq \sum_{\substack{i \in S, \text{ uncorrupted}}} (\|p_i\| + D) + \sum_{\substack{i \in S, \text{ corrupted}}} (\|q_i\| - (D + 2M)) \\
&\geq \sum_{i \in S} \|q_i\| + D \cdot \left(k - \left\lfloor \frac{k-1}{2} \right\rfloor\right) - (D + 2M) \cdot \left\lfloor \frac{k-1}{2} \right\rfloor \\
&= \sum_{i \in S} \|q_i\| + D \cdot \left(k - 2\left\lfloor \frac{k-1}{2} \right\rfloor\right) - 2M \cdot \left\lfloor \frac{k-1}{2} \right\rfloor \\
&\geq \sum_{i \in S} \|q_i\| + D - 2M \cdot \left\lfloor \frac{k-1}{2} \right\rfloor \\
&> \sum_{i \in S} \|q_i\|.
\end{aligned}$$

This contradicts the assumption that $t_m(\pi_\ell)$ minimizes $\sum_{\text{top-}k} \|q_i - \hat{s}\|$, since $0$ would be a better top-$k$-median and the optimum can only be even smaller.

Therefore it must hold that $D \leq 2M \cdot \lfloor \frac{k-1}{2} \rfloor$. Thus,

$$\sup_{\hat{\sigma}} \|t_m(\pi_\ell)\|_{\text{top-}k} \leq k \cdot (D + 2M) \leq k \cdot \left( 2M \cdot \left\lfloor \frac{k-1}{2} \right\rfloor + 2M \right) = 2M \cdot k \cdot \left\lfloor \frac{k+1}{2} \right\rfloor < \infty$$

is finite. We conclude that $\beta(t_m, \pi) \geq \lfloor \frac{k-1}{2} \rfloor + 1 = \lfloor \frac{k+1}{2} \rfloor$, as desired. $\qquad\square$

Overall, our above investigation proves the following theorem.

**Theorem A.12** (Restatement of Theorem 2.4). *Let* $\pi = (p_1, \ldots, p_m)$ *be a curve with* $p_i \in \mathbb{R}^d$. *Let* $t_m(\pi) = \sigma$ *be the curve-of-top-$k$-median. Then*

$$\beta(t_m, \pi) = \left\lfloor \frac{k+1}{2} \right\rfloor.$$

## B. Supplementary Experimental Material

All experiments were run on a HPC workstation with AMD Ryzen Threadripper PRO 5975WX, 32 cores at 3.6GHz, 512GB DDR4-3200. Our Python code is available at https://github.com/akrivosija/kDTW.

### B.1. Synthetic Data

The choice of the parameters for the curves $A_l$, $B$ and $C$, cf. Figure 1, that are used for our clustering experiment presented in Figures 2 and 5 is as follows:

- the complexity of the curves $m = 1\,001$;
- the $k$-DTW distance parameter $k = \lfloor 2.5 \cdot \log(m) \rfloor = 17$;
- the number of "peaks" in the curves in $A_l$: $l \in \{5, 6, 7, 8\}$;
- the size of the "peaks" in the $A_l$-type curves: $L = 2 \cdot \log(m) = 13.816$;
- the "small values" bound $\varepsilon = 0.2$;
- the number of curves of each type $A_l$, $B$, $C$: 20 ($n = 60$).

### B.2. Agglomerative Clustering

Clustering is an important unsupervised Machine Learning problem, that aims at partitioning the input, consisting of data objects (often simply called points), such that "similar" points are grouped in the same part, while "dissimilar" points are assigned to different parts. Center based clustering such as $k$-means require to efficiently calculate a center point for each of $k$ clusters. The input points are then assigned to the cluster represented by the closest center. Since computing a center based clustering for more than one curve is computationally hard even to approximate in most scenarios (Driemel et al., 2016; Buchin et al., 2019a; 2020; Bulteau et al., 2020), as already mentioned in the introduction, we use a popular alternative called Hierarchical Agglomerative Clustering (HAC), which requires only the pairwise distances between the input data points. The partition process starts with each curve being a singleton cluster. The current clusters are then iteratively merged in ascending order of dissimilarity. Given a dissimilarity measure $d(a, b)$ for two curves, such as Fréchet, DTW, or our novel $k$-DTW, the dissimilarity of two clusters $A$, $B$ is defined via so called *linkage functions*. The arguably most popular linkage functions are *single linkage* $d(A, B) = \min_{a \in A, b \in B} d(a, b)$ and *complete linkage* $d(A, B) = \max_{a \in A, b \in B} d(a, b)$, where the distance of two clusters is defined as the minimum resp. maximum distance of two input data points taken from each of the two clusters (Kaufman & Rousseeuw, 1990).

In Figure 5 (top) we see the results for single linkage clustering using the three distance measures. DTW (left) clearly has difficulties to distinguish between type-$A_l$ and type-$C$ curves, since the pairs $(A_l, C)$ are very close, $(A_l, B)$ are moderately close, while $(B, C)$ are far from each other, violating the triangle inequality. The Fréchet distance (right) has very large intra-cluster distances between type-$A_l$ curves that are of the same magnitude as the inter-cluster distances. With slight adjustments to the curves' parameters, we can aggravate the situation such that Fréchet cannot cluster the curves correctly either. Note, that we did not create such an extreme example as it would not only be synthetic but also *handcrafted* to a bad situation. $k$-DTW (middle) can clearly identify the clusters, as the triangle inequality is less affected, while being robust to

the spikes of type-$A_l$ curves. As a result, we obtain pure clusters whose intra-cluster distances are reasonably small and clearly distinguished from the large inter-cluster distances. The results for complete linkage are very similar, cf. Figure 5 (bottom).

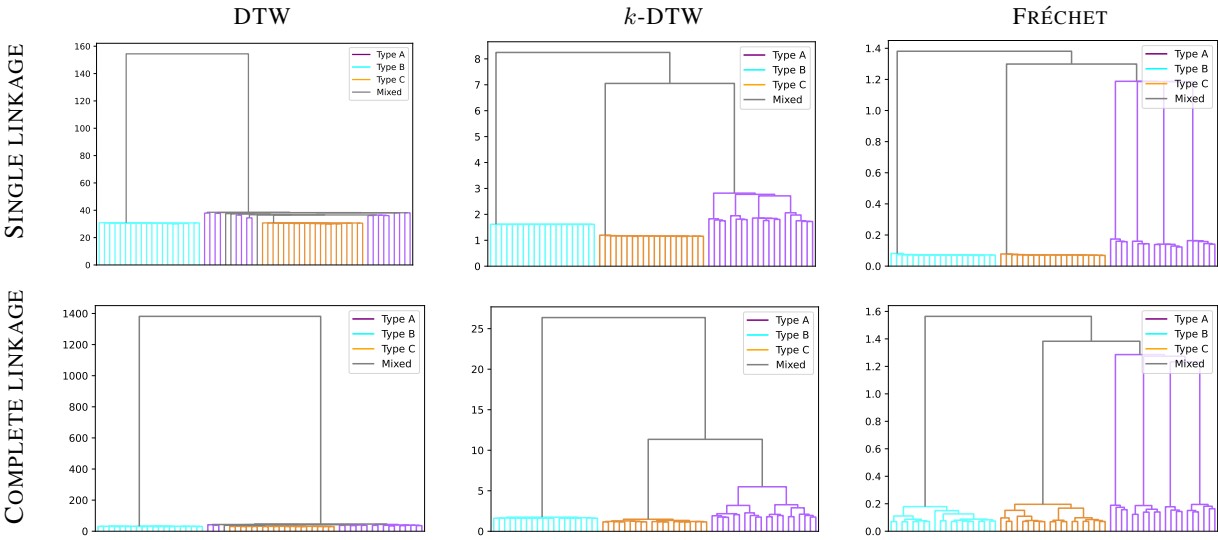

*Figure 5.* Single (top) and complete (bottom) linkage clustering; DTW (left), $k$-DTW (middle), Fréchet distance (right); synthetic data.

### B.3. $l$-Nearest Neighbor Classification

Classification is an important ML problem, where based on an input training set $X_M \subseteq X$, consisting of $M$ input points equipped with one of $K$ labels, we classify the whole domain $X$ into the $K$ categories. The $l$-nearest neighbor ($l$-NN) model provides a well-known distance based classifier (Devroye et al., 2013). Let the input training set $X_M$ consist of $M$ pairs $(\chi_i, y_i)$, where $\chi_i$ are from the instance domain $X$ with a distance measure $d : X \times X \to \mathbb{R}_{\geq 0}$, and $y_i \in [K]$, for all $i \in [M]$. Here, $X$ is the set of polygonal curves in $\mathbb{R}^d$, and the distance measures are the Fréchet, DTW, and $k$-DTW distances. Additionally, we perform experiments using several related distance measures: weak discrete Fréchet distance (Buchin et al., 2019c), edit distance with real penalty (Chen & Ng, 2004), and two variants of partial DTW distance (there are multiple distance measures under the same name in the literature). We use *partial window DTW* (Sakoe & Chiba, 1978), which matches vertices of each curve only within a frame of bounded width $w$, and *partial segment DTW* (Tak & Hwang, 2007; Luo et al., 2024), which partitions both curves into $L$ segments, each of which are matched via standard DTW. We set the window width to $w = 50$ and the number of segments to $L = 10$ in all experiments with these partial DTW variants.

Then, a curve $\tau \in X$ is classified into category $i \in [K]$, if the (relative) majority of the $l$ curves of $X_M$ that have smallest distance to $\tau$ are labeled with $y_i$ (Devroye et al., 2013).

#### B.3.1. CLASSIFICATION OF THE OPEN UNIVERSITY LEARNING ANALYTICS DATASET

We first present the results obtained on the real-world data from the Open University Learning Analytics Dataset (OULAD) (Kuzilek et al., 2017). This dataset contains data about courses, students and their interactions with a virtual learning environment. The clicks of a student, aggregated by days, are represented as clickstreams, which we interpret as polygonal curves, in order to analyze them using the Fréchet, DTW, and our novel $k$-DTW distances.

We represent $n = 275$ clickstreams of one course (a course that started in "October 2014") as polygonal curves of complexity $m = 294$. Each curve is labeled according to the achievement of the student in the course, as either "fail" (0) - 46 students, or "pass" (1) - 229 students. We use an $l$-NN model in order to predict the final exam result, which serves as a binary label indicating 'pass' or 'fail'. We note that the data set is very sparse, as it contains $56\,027$ vertices of the curves with value 0. This amounts to $69.3\%$ of all $80\,850$ vertices.

We run a 100 times repeated 6-fold cross validation, using $l$-NN with a standard choice of $l = \lceil \sqrt{n} \rceil = 17$ (cf. Devroye et al., 2013). We perform an exponential search for the best parameter $k \in \{2^i \mid i \in [8]\}$ for the $k$-DTW distance, and a

subsequent finer search. The best results were obtained for $k \in \{64, 76\}$, which amounts to roughly $20\% - 25\%$ of the curves' complexity. The results for all considered distance measures (the Fréchet distance, DTW, and $k$-DTW for various values of $k$) are given in Table 1. $k$-DTW outperformed the classification performance of Fréchet and DTW by a margin of up to $8.2\%$ resp. $1.8\%$, when measured by AUC. $k$-DTW also shows slight improvements in terms of accuracy and $F_1$-score.

*Table 2.* Binary classification of polygonal curves using $l$-nearest neighbor algorithm, on the real-world Open University Learning Analytics Dataset (OULAD) (Kuzilek et al., 2017). Mean classification performance measures (AUC, accuracy, $F_1$-score) taken over 100 independent repetitions of 6-fold cross validation and standard errors "mean (std.err.)". Best and worst values are highlighted in blue/red colors. At least one $k$-DTW variant always yields best or better while close to the best results compared to either standard DTW or Fréchet.

| Distance | AUC | Accuracy | $F_1$-Score | $T(min)$ | $T/T_{DTW}$ |
|---|---|---|---|---|---|
| Fréchet | 0.73737 (0.00194) | 0.83742 (0.00072) | 0.90953 (0.00040) | 1.7 | 1.0 |
| 2-DTW | 0.74207 (0.00197) | 0.83455 (0.00083) | 0.90779 (0.00047) | 77.4 | 47.9 |
| 4-DTW | 0.75888 (0.00184) | 0.83706 (0.00068) | 0.90929 (0.00038) | 80.8 | 50.0 |
| 8-DTW | 0.77092 (0.00164) | 0.83706 (0.00067) | 0.90946 (0.00036) | 78.5 | 48.6 |
| 16-DTW | 0.76771 (0.00164) | 0.83808 (0.00092) | 0.90932 (0.00052) | 73.7 | 45.6 |
| 32-DTW | 0.78213 (0.00160) | 0.84946 (0.00090) | 0.91438 (0.00050) | 68.9 | 42.6 |
| 56-DTW | 0.79248 (0.00158) | 0.85320 (0.00092) | 0.91651 (0.00051) | 62.8 | 38.8 |
| 60-DTW | 0.79001 (0.00157) | 0.85495 (0.00090) | 0.91760 (0.00050) | 61.7 | 38.2 |
| 64-DTW | 0.78795 (0.00150) | 0.85557 (0.00088) | 0.91796 (0.00048) | 60.6 | 37.4 |
| 68-DTW | 0.78855 (0.00156) | 0.85546 (0.00089) | 0.91790 (0.00049) | 59.5 | 36.8 |
| 72-DTW | 0.79639 (0.00152) | 0.85520 (0.00092) | 0.91775 (0.00051) | 58.3 | 36.0 |
| 76-DTW | 0.79772 (0.00150) | 0.85469 (0.00092) | 0.91744 (0.00051) | 57.6 | 35.6 |
| 80-DTW | 0.79512 (0.00150) | 0.85528 (0.00091) | 0.91779 (0.00050) | 56.3 | 34.8 |
| 84-DTW | 0.79713 (0.00145) | 0.85528 (0.00091) | 0.91779 (0.00050) | 55.2 | 34.1 |
| 88-DTW | 0.79691 (0.00147) | 0.85528 (0.00091) | 0.91779 (0.00050) | 54.2 | 33.5 |
| 92-DTW | 0.79522 (0.00151) | 0.85528 (0.00091) | 0.91779 (0.00050) | 53.2 | 32.9 |
| 96-DTW | 0.79301 (0.00147) | 0.85528 (0.00091) | 0.91779 (0.00050) | 52.3 | 32.3 |
| 100-DTW | 0.79347 (0.00142) | 0.85528 (0.00091) | 0.91779 (0.00050) | 51.3 | 31.7 |
| 104-DTW | 0.79191 (0.00147) | 0.85528 (0.00091) | 0.91779 (0.00050) | 50.5 | 31.2 |
| 108-DTW | 0.78859 (0.00146) | 0.85528 (0.00091) | 0.91779 (0.00050) | 49.7 | 30.7 |
| 112-DTW | 0.78556 (0.00148) | 0.85528 (0.00091) | 0.91779 (0.00050) | 49.0 | 30.3 |
| 116-DTW | 0.78606 (0.00150) | 0.85528 (0.00091) | 0.91779 (0.00050) | 48.1 | 29.7 |
| 120-DTW | 0.78541 (0.00152) | 0.85528 (0.00091) | 0.91779 (0.00050) | 47.5 | 29.4 |
| 124-DTW | 0.78399 (0.00155) | 0.85528 (0.00091) | 0.91779 (0.00050) | 46.8 | 28.9 |
| 128-DTW | 0.78620 (0.00159) | 0.85528 (0.00091) | 0.91779 (0.00050) | 46.4 | 28.7 |
| 256-DTW | 0.77699 (0.00164) | 0.85528 (0.00091) | 0.91779 (0.00050) | 34.1 | 21.0 |
| DTW | 0.78360 (0.00160) | 0.85459 (0.00085) | 0.91735 (0.00047) | 1.6 | 1.0 |
| EditRealPenalty | 0.65800 (0.00261) | 0.79892 (0.00335) | 0.86557 (0.00348) | 4.0 | 2.5 |
| PartSegmDTW | 0.54652 (0.00255) | 0.79778 (0.00180) | 0.88161 (0.00149) | 0.2 | 0.1 |
| PartWinDTW | 0.77959 (0.00175) | 0.87823 (0.00051) | 0.92956 (0.00031) | 0.5 | 0.3 |
| WeakFréchet | 0.73917 (0.00204) | 0.83618 (0.00073) | 0.90896 (0.00041) | 3.0 | 1.9 |

### B.3.2. PARAMETER $k$ TUNING AND EVALUATION ON HOLD OUT DATA

In addition to our explicit parameter search, cross validated over the full data, we perform another hold out evaluation. Here the parameter $k$ of the $k$-DTW distance is cross validated on training data, and the best value of $k$ is then used on a hold out test data set, for $l$-nearest neighbor classification. For that sake, we split the input into training and test sets $A$ and $B$, respectively. We randomly split the OULAD dataset (Kuzilek et al., 2017), where the test set $B$ has a $\{\frac{1}{3}, \frac{1}{4}, \frac{1}{5}\}$ fraction of the input. We find the best value of $k$ on $A$ and evaluate it on $B$. Finally, we compare the performances of the Fréchet, $k$-DTW and DTW distances on the test set $B$. In all experiments we run 100 times repeated 6-fold cross-validation, using $l = \lceil \sqrt{|A|} \rceil$ for the $l$-NN training and test on the remaining hold out data.

The results are presented in Tables 3 to 5. In all of the experiments, the best $k$-DTW variant that is selected on the training set is either best or close to the best, compared to the standard DTW and the Fréchet distance. Note that very small choices of the split size imply that the classification quality on the test set significantly decreases (e.g. in Table 5).

*Table 3.* Binary classification of polygonal curves using $l$-nearest neighbor algorithm, on the real-world OULAD dataset (Kuzilek et al., 2017). Training set consists of $\frac{2}{3}$ of the input set. Testing set is remaining $\frac{1}{3}$. Mean classification performance measures (AUC, accuracy, $F_1$-score) taken over 100 independent repetitions of 6-fold cross validation and standard errors "mean (std.err.)". Best values are highlighted in blue color.

| | | | Training | | | |
|---|---|---|---|---|---|---|
| **Distance** | **AUC** | **Accuracy** | $F_1$-**Score** | **Distance** | **AUC** | **Accuracy** | $F_1$-**Score** |
| 2-DTW | 0.71501 | 0.84599 | 0.91524 | 88-DTW | 0.75375 | 0.85425 | 0.91969 |
| 4-DTW | 0.70214 | 0.84653 | 0.91554 | 92-DTW | 0.74882 | 0.85425 | 0.91969 |
| 8-DTW | 0.70701 | 0.84845 | 0.91661 | 96-DTW | 0.75097 | 0.85425 | 0.91969 |
| 16-DTW | 0.69618 | 0.85288 | 0.91907 | 100-DTW | 0.74912 | 0.85425 | 0.91969 |
| 32-DTW | 0.71676 | 0.84858 | 0.91651 | 104-DTW | 0.74578 | 0.85425 | 0.91969 |
| 56-DTW | 0.74510 | 0.85508 | 0.92015 | 108-DTW | 0.74237 | 0.85425 | 0.91969 |
| 60-DTW | 0.75037 | 0.85524 | 0.92024 | 112-DTW | 0.74483 | 0.85425 | 0.91969 |
| 64-DTW | 0.75410 | 0.85192 | 0.91832 | 116-DTW | 0.74176 | 0.85425 | 0.91969 |
| 68-DTW | 0.76162 | 0.85431 | 0.91971 | 120-DTW | 0.74261 | 0.85425 | 0.91969 |
| 72-DTW | 0.76220 | 0.85437 | 0.91975 | 124-DTW | 0.74078 | 0.85425 | 0.91969 |
| 76-DTW | 0.75543 | 0.85426 | 0.91969 | 128-DTW | 0.73960 | 0.85425 | 0.91969 |
| 80-DTW | 0.75682 | 0.85404 | 0.91957 | 256-DTW | 0.72549 | 0.85425 | 0.91969 |
| 84-DTW | 0.75366 | 0.85425 | 0.91969 | | | | |

| | Testing | | |
|---|---|---|---|
| **Distance** | **AUC** | **Accuracy** | $F_1$-**Score** |
| 72-DTW | 0.78912 (0.00382) | 0.80103 (0.00266) | 0.87266 (0.00174) |
| DTW | 0.77772 (0.00381) | 0.79724 (0.00258) | 0.87005 (0.00167) |
| Fréchet | 0.65995 (0.00542) | 0.78542 (0.00144) | 0.87542 (0.00087) |

*Table 4.* Binary classification of polygonal curves using $l$-nearest neighbor algorithm, on the real-world OULAD dataset (Kuzilek et al., 2017). Training set consists of $\frac{3}{4}$ of the input set. Testing set is remaining $\frac{1}{4}$. Mean classification performance measures (AUC, accuracy, $F_1$-score) taken over 100 independent repetitions of 6-fold cross validation and standard errors "mean (std.err.)". Best values are highlighted in blue color.

| | | | | Training | | | |
|---|---|---|---|---|---|---|---|
| **Distance** | **AUC** | **Accuracy** | **$F_1$-Score** | **Distance** | **AUC** | **Accuracy** | **$F_1$-Score** |
| 2-DTW | 0.74698 | 0.83849 | 0.91098 | 88-DTW | 0.77448 | 0.84538 | 0.91465 |
| 4-DTW | 0.73626 | 0.83941 | 0.91141 | 92-DTW | 0.77248 | 0.84538 | 0.91465 |
| 8-DTW | 0.74288 | 0.83839 | 0.91089 | 96-DTW | 0.77127 | 0.84538 | 0.91465 |
| 16-DTW | 0.74744 | 0.83912 | 0.91130 | 100-DTW | 0.77002 | 0.84538 | 0.91465 |
| 32-DTW | 0.75834 | 0.83714 | 0.90998 | 104-DTW | 0.76765 | 0.84538 | 0.91465 |
| 56-DTW | 0.77456 | 0.84528 | 0.91459 | 108-DTW | 0.76388 | 0.84538 | 0.91465 |
| 60-DTW | 0.78048 | 0.84533 | 0.91462 | 112-DTW | 0.76606 | 0.84538 | 0.91465 |
| 64-DTW | 0.78281 | 0.84426 | 0.91401 | 116-DTW | 0.76558 | 0.84538 | 0.91465 |
| 68-DTW | 0.78641 | 0.84538 | 0.91465 | 120-DTW | 0.76449 | 0.84538 | 0.91465 |
| 72-DTW | 0.78423 | 0.84538 | 0.91465 | 124-DTW | 0.76259 | 0.84538 | 0.91465 |
| 76-DTW | 0.78071 | 0.84538 | 0.91465 | 128-DTW | 0.76108 | 0.84538 | 0.91465 |
| 80-DTW | 0.78088 | 0.84513 | 0.91451 | 256-DTW | 0.75135 | 0.84538 | 0.91465 |
| 84-DTW | 0.77768 | 0.84538 | 0.91465 | | | | |

| | Testing | | |
|---|---|---|---|
| **Distance** | **AUC** | **Accuracy** | **$F_1$-Score** |
| 68-DTW | 0.74092 (0.00533) | 0.79057 (0.00164) | 0.87842 (0.00102) |
| DTW | 0.73140 (0.00583) | 0.79001 (0.00159) | 0.87812 (0.00098) |
| Fréchet | 0.57310 (0.00665) | 0.79696 (0.00022) | 0.88270 (0.00028) |

*Table 5.* Binary classification of polygonal curves using $l$-nearest neighbor algorithm, on the real-world OULAD dataset (Kuzilek et al., 2017). Training set consists of $\frac{4}{5}$ of the input set. Testing set is remaining $\frac{1}{5}$. Mean classification performance measures (AUC, accuracy, $F_1$-score) taken over 100 independent repetitions of 6-fold cross validation and standard errors "mean (std.err.)". Best values are highlighted in blue color.

| | | | | Training | | | |
|---|---|---|---|---|---|---|---|
| **Distance** | **AUC** | **Accuracy** | **$F_1$-Score** | **Distance** | **AUC** | **Accuracy** | **$F_1$-Score** |
| 2-DTW | 0.77194 | 0.85320 | 0.91767 | 88-DTW | 0.81430 | 0.85380 | 0.91729 |
| 4-DTW | 0.77292 | 0.85021 | 0.91620 | 92-DTW | 0.81094 | 0.85380 | 0.91729 |
| 8-DTW | 0.76751 | 0.84344 | 0.91276 | 96-DTW | 0.81126 | 0.85380 | 0.91729 |
| 16-DTW | 0.77602 | 0.84038 | 0.91079 | 100-DTW | 0.81037 | 0.85380 | 0.91729 |
| 32-DTW | 0.79556 | 0.84879 | 0.91431 | 104-DTW | 0.80934 | 0.85380 | 0.91729 |
| 56-DTW | 0.81157 | 0.85394 | 0.91737 | 108-DTW | 0.80691 | 0.85380 | 0.91729 |
| 60-DTW | 0.81480 | 0.85394 | 0.91737 | 112-DTW | 0.80950 | 0.85380 | 0.91729 |
| 64-DTW | 0.81617 | 0.85348 | 0.91711 | 116-DTW | 0.80972 | 0.85380 | 0.91729 |
| 68-DTW | 0.81908 | 0.85384 | 0.91732 | 120-DTW | 0.80900 | 0.85380 | 0.91729 |
| 72-DTW | 0.82163 | 0.85375 | 0.91726 | 124-DTW | 0.80801 | 0.85380 | 0.91729 |
| 76-DTW | 0.81766 | 0.85371 | 0.91724 | 128-DTW | 0.80602 | 0.85380 | 0.91729 |
| 80-DTW | 0.81915 | 0.85357 | 0.91716 | 256-DTW | 0.79856 | 0.85380 | 0.91729 |
| 84-DTW | 0.81579 | 0.85380 | 0.91729 | | | | |

| | Testing | | |
|---|---|---|---|
| **Distance** | **AUC** | **Accuracy** | **$F_1$-Score** |
| 72-DTW | 0.63990 (0.00602) | 0.79996 (0.00021) | 0.88358 (0.00031) |
| DTW | 0.62175 (0.00675) | 0.79996 (0.00021) | 0.88358 (0.00031) |
| Fréchet | 0.48112 (0.00817) | 0.79996 (0.00021) | 0.88358 (0.00031) |

### B.3.3. CLASSIFICATION OF FURTHER DATASETS

We have evaluated the $l$-Nearest Neighbor Classification on additional datasets that have been used for evaluation of different distance measures in (Aghababa & Phillips, 2023). Their $l$-NN experiments are performed with $l = 5$ and evaluated on a random 70/30 hold-out split, while we choose $l = \lceil \sqrt{n} \rceil$, as for the previous OULAD (Kuzilek et al., 2017) dataset and evaluate by a 6-fold cross validation, 100 times repeated independently. We compare $k$-DTW again against the discrete Fréchet and the DTW distance, as in the rest of our manuscript.

Here, we need to choose the value of $k$ for our $k$-DTW distance. We will see that it is not needed to perform the extensive exponential or linear search, as in practice it suffices to choose some small value, compared to the input curves' complexity $m$. Thus, we compare the $k$-DTW for the values of $k \in \{\ln(m), \sqrt{m}, m/10, m/4\}$, and we stress that the linear values are only needed for short curves which was underlined by our theoretical results. Note that in the paper of Aghababa & Phillips (2023) there is no analysis of the complexity of the input curves $m$ – we present these values, together with further descriptive parameters of the observed real-world datasets in Table 6.

*Table 6.* Description of the real-world datasets from (Aghababa & Phillips, 2023), where each curve has a label from $\{0, 1\}$. $n$ – the number of the curves in a dataset. $m$ – the complexity of curves in a dataset. $d$ – the dimension of the ambient space.

| Dataset | Reference | # Curves | Categories | Min | Max | Average | Dim. |
|---|---|---|---|---|---|---|---|
| | | $n$ | | $m$ | $m$ | $m$ | $d$ |
| Cars+Bus | (Cruz et al., 2016) | 120 | $(76, 44)$ | 10 | 645 | 146 | 2 |
| Sim. C+B | (Aghababa & Phillips, 2023) | 446 | $(226, 220)$ | 9 | 645 | 126 | 2 |
| Char0uw | (Williams, 2008) | 256 | $(131, 125)$ | 89 | 153 | 117 | 2 |
| Char1nw | (Williams, 2008) | 255 | $(130, 125)$ | 108 | 143 | 126 | 2 |
| Char2nu | (Williams, 2008) | 261 | $(130, 131)$ | 92 | 182 | 120 | 2 |
| Two Persons | (UIC, 2006) | 213 | $(124, 89)$ | 72 | 5 493 | 1 175 | 2 |

The dataset "Simulated C+B" (Aghababa & Phillips, 2023) was obtained from the dataset "Cars+Bus" (Cruz et al., 2016), to balance the number of input curves of the two classes by adding random noise to original input curves. For both the task is to distinguish between paths that are driven by car vs. bus. The "Characters" dataset (Williams, 2008) contains multiple hand-written characters, from which we picked three exemplary and challenging pairs $(u,w)$, $(n,w)$ and $(n,u)$ to be distinguished. Datasets are denoted here as "Char0uw", "Char1nw" and "Char2nu", respectively. The "Two Persons" (UIC, 2006) dataset contains trajectories walked by two different persons to be distinguished. As it was done in (Aghababa & Phillips, 2023) for all datasets, we also removed stationary points along the trajectories, as this does not change the shape of a polygonal curve.

Tables 7 and 8 show the classification performance results: the means and standard errors over 100 independent repetitions. Additionally, we present the total running times for all experiments, and the ratio to the running time of the corresponding experiment using the DTW distance.

The best performing measure for each case is highlighted in blue, and the worst in red. One can see that in the majority of cases, $k$-DTW either still performs best, or close to the best. The only exception is "Char0uw" where ERP excels and beats **all** others by a large margin. $k$-DTW is still second best in these cases. Notably, **all** competitors have some worst cases, while $k$-DTW is **never** worst.

*Table 7.* Mean classification performance measures (AUC, accuracy, $F_1$-score) taken over 100 independent repetitions of 6-fold cross validation and standard errors "mean (std.err.)". Best and worst values are highlighted in blue/red colors. Note that in half of the combinations of dataset/performance measure, some $k$-DTW variant is best, while worst results are mostly attained using standard DTW or Fréchet. At least one $k$-DTW variant always yields best or better while close to the best results compared to either standard DTW or Fréchet. Datasets taken from (Aghababa & Phillips, 2023). Running time ($T$) given in minutes.

| Data | Distance | AUC | Accuracy | $F_1$-Score | $T$(min) | $T/T_{DTW}$ |
|---|---|---|---|---|---|---|
| Cars+Bus | Fréchet | 0.51511 (0.00390) | 0.53158 (0.00355) | 0.64938 (0.00299) | 1.1 | 1.0 |
| | $\ln(m)$-DTW | 0.54830 (0.00383) | 0.56192 (0.00308) | 0.67068 (0.00252) | 5.3 | 5.0 |
| | $\sqrt{m}$-DTW | 0.52259 (0.00369) | 0.54058 (0.00280) | 0.64863 (0.00261) | 5.7 | 5.4 |
| | $m/10$-DTW | 0.53635 (0.00342) | 0.53008 (0.00323) | 0.62415 (0.00301) | 8.1 | 7.6 |
| | $m/4$-DTW | 0.57633 (0.00329) | 0.56250 (0.00266) | 0.64711 (0.00264) | 16.5 | 15.5 |
| | DTW | 0.57590 (0.00321) | 0.55417 (0.00273) | 0.63637 (0.00269) | 1.1 | 1.0 |
| | EditRealPenalty | 0.56580 (0.00324) | 0.54900 (0.00258) | 0.62282 (0.00269) | 2.4 | 2.2 |
| | PartSegmDTW | 0.54492 (0.00358) | 0.58108 (0.00293) | 0.70430 (0.00236) | 0.2 | 0.1 |
| | PartWinDTW | 0.58362 (0.00383) | 0.60033 (0.00314) | 0.71338 (0.00258) | 0.8 | 0.7 |
| | WeakFréchet | 0.51607 (0.00396) | 0.53208 (0.00346) | 0.64876 (0.00298) | 1.7 | 1.6 |
| Sim. C+B | Fréchet | 0.88027 (0.00049) | 0.77717 (0.00075) | 0.76358 (0.00091) | 25.0 | 1.1 |
| | $\ln(m)$-DTW | 0.88748 (0.00046) | 0.77291 (0.00074) | 0.75119 (0.00092) | 61.6 | 2.8 |
| | $\sqrt{m}$-DTW | 0.88966 (0.00045) | 0.76037 (0.00073) | 0.73046 (0.00098) | 69.5 | 3.1 |
| | $m/10$-DTW | 0.89769 (0.00046) | 0.76860 (0.00077) | 0.72968 (0.00093) | 97.4 | 4.4 |
| | $m/4$-DTW | 0.91712 (0.00041) | 0.79836 (0.00052) | 0.76566 (0.00076) | 158.5 | 7.1 |
| | DTW | 0.91829 (0.00038) | 0.79339 (0.00076) | 0.75607 (0.00110) | 22.3 | 1.0 |
| | EditRealPenalty | 0.90888 (0.00041) | 0.79709 (0.00059) | 0.76120 (0.00088) | 50.6 | 2.3 |
| | PartSegmDTW | 0.80593 (0.00088) | 0.58766 (0.00071) | 0.31972 (0.00168) | 3.6 | 0.2 |
| | PartWinDTW | 0.83221 (0.00068) | 0.63397 (0.00092) | 0.46484 (0.00178) | 14.8 | 0.7 |
| | WeakFréchet | 0.87942 (0.00047) | 0.77073 (0.00071) | 0.75483 (0.00089) | 37.6 | 1.7 |
| Char0uw | Fréchet | 0.98094 (0.00029) | 0.93426 (0.00053) | 0.93568 (0.00056) | 0.9 | 0.9 |
| | $\ln(m)$-DTW | 0.98374 (0.00025) | 0.93341 (0.00061) | 0.93553 (0.00061) | 12.5 | 11.9 |
| | $\sqrt{m}$-DTW | 0.98434 (0.00026) | 0.93139 (0.00064) | 0.93426 (0.00063) | 14.9 | 14.1 |
| | $m/10$-DTW | 0.98433 (0.00028) | 0.92863 (0.00050) | 0.93181 (0.00052) | 15.8 | 15.0 |
| | $m/4$-DTW | 0.98474 (0.00024) | 0.92652 (0.00058) | 0.93020 (0.00058) | 21.1 | 20.1 |
| | DTW | 0.98625 (0.00028) | 0.91586 (0.00066) | 0.92050 (0.00064) | 1.1 | 1.0 |
| | EditRealPenalty | 0.99819 (0.00009) | 0.97906 (0.00025) | 0.97930 (0.00029) | 2.3 | 2.2 |
| | PartSegmDTW | 0.99712 (0.00016) | 0.97793 (0.00029) | 0.97840 (0.00033) | 0.1 | 0.1 |
| | PartWinDTW | 0.98622 (0.00027) | 0.91445 (0.00059) | 0.91930 (0.00059) | 0.7 | 0.6 |
| | WeakFréchet | 0.97652 (0.00033) | 0.92185 (0.00072) | 0.92257 (0.00075) | 1.2 | 1.1 |

*Table 8.* Mean classification performance measures (AUC, accuracy, $F_1$-score) taken over 100 independent repetitions of 6-fold cross validation and standard errors "mean (std.err.)". Best and worst values are highlighted in blue/red colors. Note that in half of the combinations of dataset/performance measure, some $k$-DTW variant is best, while worst results are mostly attained using standard DTW or Fréchet. At least one $k$-DTW variant always yields best or better while close to the best results compared to either standard DTW or Fréchet. Datasets taken from (Aghababa & Phillips, 2023). Running time ($T$) given in minutes.

| Data | Distance | AUC | Accuracy | $F_1$-Score | $T$(min) | $T/T_{DTW}$ |
|------|----------|-----|----------|-------------|----------|-------------|
| Char1nw | Fréchet | 0.91701 (0.00080) | 0.83265 (0.00130) | 0.82623 (0.00150) | 3.3 | 1.1 |
| | $\ln(m)$-DTW | 0.92853 (0.00063) | 0.84517 (0.00114) | 0.84209 (0.00120) | 22.5 | 7.4 |
| | $\sqrt{m}$-DTW | 0.94034 (0.00057) | 0.86560 (0.00107) | 0.86518 (0.00113) | 30.4 | 10.0 |
| | $m/10$-DTW | 0.94180 (0.00054) | 0.86354 (0.00125) | 0.86334 (0.00121) | 39.0 | 12.9 |
| | $m/4$-DTW | 0.95289 (0.00045) | 0.88475 (0.00111) | 0.88545 (0.00113) | 45.1 | 14.9 |
| | DTW | 0.95115 (0.00047) | 0.87420 (0.00113) | 0.87720 (0.00105) | 3.0 | 1.0 |
| | EditRealPenalty | 0.95442 (0.00048) | 0.86093 (0.00100) | 0.85042 (0.00121) | 6.9 | 2.3 |
| | PartSegmDTW | 0.95931 (0.00042) | 0.88577 (0.00089) | 0.87933 (0.00099) | 0.4 | 0.1 |
| | PartWinDTW | 0.94604 (0.00051) | 0.87546 (0.00109) | 0.87802 (0.00107) | 2.0 | 0.7 |
| | WeakFréchet | 0.91767 (0.00083) | 0.83866 (0.00139) | 0.82806 (0.00164) | 3.8 | 1.2 |
| Char2nu | Fréchet | 0.98326 (0.00026) | 0.93725 (0.00074) | 0.93259 (0.00084) | 4.7 | 0.9 |
| | $\ln(m)$-DTW | 0.98517 (0.00025) | 0.94303 (0.00062) | 0.93888 (0.00069) | 23.2 | 4.3 |
| | $\sqrt{m}$-DTW | 0.98615 (0.00023) | 0.94415 (0.00049) | 0.94067 (0.00057) | 30.0 | 5.5 |
| | $m/10$-DTW | 0.98816 (0.00018) | 0.94228 (0.00053) | 0.93861 (0.00063) | 38.9 | 7.2 |
| | $m/4$-DTW | 0.98949 (0.00016) | 0.94757 (0.00048) | 0.94475 (0.00057) | 44.6 | 8.2 |
| | DTW | 0.99376 (0.00019) | 0.95534 (0.00045) | 0.95348 (0.00048) | 5.4 | 1.0 |
| | EditRealPenalty | 0.99830 (0.00007) | 0.96695 (0.00030) | 0.96573 (0.00035) | 12.3 | 2.3 |
| | PartSegmDTW | 0.99684 (0.00011) | 0.96116 (0.00038) | 0.95963 (0.00043) | 0.7 | 0.1 |
| | PartWinDTW | 0.99355 (0.00019) | 0.95519 (0.00045) | 0.95334 (0.00049) | 3.5 | 0.6 |
| | WeakFréchet | 0.98281 (0.00025) | 0.93364 (0.00066) | 0.92836 (0.00075) | 6.5 | 1.2 |
| Two Persons | Fréchet | 0.95373 (0.00058) | 0.94832 (0.00002) | 0.95324 (0.00021) | 57.3 | 0.9 |
| | $\ln(m)$-DTW | 0.95630 (0.00059) | 0.94832 (0.00002) | 0.95324 (0.00021) | 655.2 | 10.4 |
| | $\sqrt{m}$-DTW | 0.95411 (0.00057) | 0.94832 (0.00002) | 0.95324 (0.00021) | 962.6 | 15.3 |
| | $m/10$-DTW | 0.96191 (0.00053) | 0.94832 (0.00002) | 0.95324 (0.00021) | 1570.8 | 25.0 |
| | $m/4$-DTW | 0.96426 (0.00055) | 0.94390 (0.00012) | 0.94939 (0.00025) | 2857.9 | 45.6 |
| | DTW | 0.96115 (0.00053) | 0.94832 (0.00002) | 0.95324 (0.00021) | 62.7 | 1.0 |
| | EditRealPenalty | 0.85989 (0.00115) | 0.77931 (0.00004) | 0.76642 (0.00051) | 130.5 | 2.1 |
| | PartSegmDTW | 0.95432 (0.00060) | 0.90336 (0.00045) | 0.90999 (0.00051) | 6.1 | 0.1 |
| | PartWinDTW | 0.94616 (0.00052) | 0.91961 (0.00026) | 0.92637 (0.00037) | 35.0 | 0.6 |
| | WeakFréchet | 0.95376 (0.00056) | 0.94832 (0.00002) | 0.95324 (0.00021) | 92.7 | 1.5 |

