# OpenReview forum: "Improved Learning via k-DTW: A Novel Dissimilarity Measure for Curves"
_ICML.cc/2025/Conference — ICML 2025 poster_

### Official Review · Reviewer_VKu5 · 2025-03-09

**Overall Recommendation:** 4

**Summary:**

The paper proposes k-DTW (k-Dynamic Time Warping) as a dissimilarity measure between polygonal curves. The motivation is that many diverse datasets can be thought of as curves and measuring appropriately the distance between them is a fundamental problem.  Usually researchers has studied Frechet distances and DTW for this purpose and typically some transformations of one
curve into another is involved. However, both aforementioned measures have their disadvantages: Frechet distance is very sensitive to outlier whereas DTW is not a metric since it does not satisfy the triangle inequality.

To alleviate these issues with existing distances, the paper proposes k-DTW which combines advantages of standard measures like Frechet distance and of the standard DTW (essentially interpolating between the two), without their disadvantages.

At a high-level, the parameter k controls the degree of accuracy for the transformation performed between the two curves. Oftentimes, carrying out the whole trasformation may be expensive and also overfit to noise, whereas k-DTW only cares about the most important parts of the trasformation, as measured by a small subset of size k, and ignores the rest of the transformation. Although this may seem lossy it offers some advantages.

The main contributions of the paper are:

1) The first point the authors make is that k-DTW satisfies a strengthened triangle inequality compared to DTW and it is thus closer to a proper metric, while retaining some robustness of DTW.

 2) The second point is that there is an exact algorithm, as well as a (1 + eps)-approximation for k-DTW using a parametric search for the k-th largest matched distance with standard DTW on modified distance matrices as a subroutine.

3) Next, they show the first dimension-free learning bounds for clustering under k-DTW and a separation result showing that k-DTW has strictly smaller Rademacher and Gaussian complexity than DTW for clustering curves.

4) Finally, the authors provide experiments showing the benefits of k-DTW over the other two measures in the setting of clustering and nearest neighbor classification, on synthetic and real world data.

**Claims And Evidence:**

yes

**Essential References Not Discussed:**

no

**Experimental Designs Or Analyses:**

yes.

**Methods And Evaluation Criteria:**

yes

**Other Comments Or Suggestions:**

-

**Other Strengths And Weaknesses:**

The conceptual message of the paper is that if we don't care about the whole computation of DTW or Frechet and we rather focus on the most important k parts of the computation then we can have good algorithms. The idea is that curves can be approximated by polygonal curves if we select some vertices on them and we connect them with affine segments, essentially interpolating between the points on the curves.

The main definition introduced in the paper is Definition 2.2 with a parameter k: as k increases we care about more and more pairs of large distances between the two curves; specifically, for k = 1 we recover the Frechet distance, and for k large
enough we recover the DTW distance.

I find the definition here very natural, and it comes as no surprise that once we focus on k terms in the summation, then then triangle inequality will be respected up to a multiplicative factor of k (Lemma 2.3).

The authors after introducing their k-DTW distance, they rule out some perhaps straighforward approaches. One thing that is notable is that the k-DTW distancε between two curves is NOT equal to just taking the largest k distances from the sum that yields their DTW distance.

Their first main result is to give an exact algorithm with runtime proportional to the description of the curves and the number of distinct pairwise distances. The analysis is non-trivial but it follows from the top-k framework of (Bertsimas & Sim, 2003).

The second main result is a (1+eps)-aox with a better runtime. The authors essentially shave off the number of distinct pairwise distances and replace it with the 1/eps x logarithm of k (say the accuracy eps is fixed small number).

The third main result, is what happens when we want to learn the median of curves sampled from a distribution over curves with vertices in the unit Euclidean ball of dimension d. The authors calculate the Rademacher and Gaussian complexities both for DTW and for k-DTW and they show how they can replace a strong dependence on the size of the polygonal curve with their parameter k.  Importantly, because k can be much smaller than dimension d, this provides a dimension-independent bound for learning.

No major weaknesses could be found.

**Questions For Authors:**

-

**Relation To Broader Scientific Literature:**

Important relation since k-DTW could improve efficiency for comparing curves.

**Theoretical Claims:**

yes, except the learning section 4.

---

> ### Author Rebuttal · Authors · 2025-03-31
>
> We thank the reviewer for the valuable feedback.

---

### Official Review · Reviewer_uWfM · 2025-03-12

**Overall Recommendation:** 4

**Summary:**

The paper proposes a new distance for between curves, the k-DTW distance, and make several types of compelling arguments for its use.  These include better near-metric properties than DTW, and better learning complexity than Frechet distance - k-DTW generalizes both.  The paper provides an efficient approximation algorithm (with clear provable guarantees), heuristics for improving runtime in practice while maintaining guarantees, and learning complexity results for learning a median curve from a set.  Each step involves non-trivial insights.  Moreover, the paper shows empirical improvement on synthetic (for clustering) and real data sets (for classification) that demonstrate clear improvement over the standard DTW and Frechet distances.

Data analysis on curve data is an active area, and this paper provides an inventive and comprehensively presented new method in this area.  Moreover, I found it surprising that the new method works so well - it considers the longest k distances in a matched monotonic sequence (DTW uses all, and Frechet uses the 1 longest), and this has very different properties and seems to be more robust than either.

Moreover I found the paper clearly written.  It combines a variety of complicated theoretical and practical perspectives.

**Claims And Evidence:**

Yes

**Essential References Not Discussed:**

No

**Experimental Designs Or Analyses:**

They seem reasonable.  There are three experiments: clustering (synthetic -- it illustrates the advantages of k-DTW well), classification on learning analytics (theirs does best among DTW, Frechet), and classification letters/trajectories (theirs often does best).

I would have liked to have seen a more explicit parameter search over k on the training data, and then used once on evaluation data.

**Methods And Evaluation Criteria:**

Yes

**Other Comments Or Suggestions:**

N/A

**Other Strengths And Weaknesses:**

N/A

**Questions For Authors:**

N/A

**Relation To Broader Scientific Literature:**

I think it is fine.  It covers important aspects of curve distances, although it may be useful to compare to things like Edit Distance with Real Penalties:
  Chen and Ng. "On the marriage of lp-norms and edit distance" in VLDB 2004

**Theoretical Claims:**

Yes (for the most part), the properties proposed to justify the metric are clearly explained and stated formally.  The main proofs are proven or sketched in the main paper, and the ones deferred to the appendix make sense, and I did not find any concerns.

---

> ### Author Rebuttal · Authors · 2025-03-31
>
> We thank the reviewer for the valuable feedback.
>
> **Question 1:** Edit Distance with Real Penalties, as in: Chen and Ng, VLDB 2004.
>
> **Answer 1:**  We will include some references, discussions and baseline experiments for your suggestion along with another suggestion of reviewer "MvAs". Please see our reply to "MvAs" for details on the other distance measures.
>
> The following table shows the results of our experiments, extended to weak Fréchet distance, partial DTW, and Edit Distance with Real Penalties, accompanied with the best performing $k$-DTW (cf. Table 4 in our submission).
>
> The best performing measure for each case is highlighted in $\textcolor{blue}{\text{blue}}$, and the worst in $\textcolor{red}{\text{red}}$. One can see that in the majority of cases, $k$-DTW either still performs $\textcolor{blue}{\text{best}}$, or $\textcolor{orange}{\text{close to the best}}$. The only exception is "Char0uw" where ERP excels and beats *all* others by a large margin. $k$-DTW is still second best in these cases. Notably, *all* competitors have some worst cases, while $k$-DTW is *never* worst.
>
> | Dataset    | Distance        | AUC (std.err.)| Accuracy (std.err.)       | $F_1$-Score (std.err.)    |
> |------------|-----------------|---------------------------------------|---------------------------------------|---------------------------------------|
> | Cars+Bus   | PartialDTW      | $\textcolor{blue}{0.58362 (0.00383)}$     | $\textcolor{blue}{0.60033 (0.00314)}$   | $\textcolor{blue}{0.71338 (0.00258)}$  |
> || WeakFréchet   | $\textcolor{red}{0.51607 (0.00396)}$    | $\textcolor{red}{0.53208 (0.00346)}$    | ${0.64876 (0.00298)}$       |
> || EditRealPenalty | $0.56580 (0.00324)$         | $0.54900 (0.00258)$         | $\textcolor{red}{0.62282 (0.00269)}$    |
> || $m/4$-DTW       | $\textcolor{orange}{0.57633 (0.00329)}$ | $\textcolor{orange}{0.56250 (0.00266)}$ | $0.64711 (0.00264)$         |
> |
> | Sim.C+B    | PartialDTW      | $\textcolor{red}{0.83221 (0.00068)}$    | $\textcolor{red}{0.63397 (0.00092)}$    | $\textcolor{red}{0.46484 (0.00178)}$    |
> || WeakFréchet   | $0.87942 (0.00047)$         | $0.77073 (0.00071)$         | $0.75483 (0.00089)$         |
> || EditRealPenalty | ${0.90888 (0.00041)}$       | ${0.79709 (0.00059)}$       | ${0.76120 (0.00088)}$       |
> || $m/4$-DTW       | $\textcolor{blue}{0.91712 (0.00041)}$   | $\textcolor{blue}{0.79836 (0.00052)}$   | $\textcolor{blue}{0.76566 (0.00076)}$   |
> |
> | Char0uw    | PartialDTW      | $0.98622 (0.00027)$         | $\textcolor{red}{0.91445 (0.00059)}$    | $\textcolor{red}{0.91930 (0.00059)}$    |
> || WeakFréchet   | $\textcolor{red}{0.97652 (0.00033)}$    | $0.92185 (0.00072)$         | $0.92257	(0.00075)$         |
> || EditRealPenalty | $\textcolor{blue}{0.99819 (0.00009)}$   | $\textcolor{blue}{0.97906 (0.00025)}$   | $\textcolor{blue}{0.97930 (0.00029)}$   |
> || $\ln(m)$-DTW    | $\textcolor{orange}{0.98374 (0.00025)}$ | $0.93341 (0.00061)$         | $0.93553 (0.00061)$        |
> |
> | Char1nw    | PartialDTW      | $0.94604 (0.00051)$         | ${0.87546 (0.00109)}$       | ${0.87802 (0.00107)}$       |
> || WeakFréchet   | $\textcolor{red}{0.91767 (0.00083)}$    | $\textcolor{red}{0.83866 (0.00139)}$    | $\textcolor{red}{0.82806 (0.00164)}$    |
> || EditRealPenalty | $\textcolor{blue}{0.95442 (0.00048)}$   | $0.86093 (0.00100)$         | $0.85042 (0.00121)$         |
> || $m/4$-DTW       | $\textcolor{orange}{0.95289 (0.00045)}$ | $\textcolor{blue}{0.88475 (0.00111)}$   | $\textcolor{blue}{0.88545 (0.00113)}$   |
> |
> | Char2nu    | PartialDTW      | ${0.99355 (0.00019)} $      | ${0.95519 (0.00045)}$       | ${0.95334 (0.00049)}$      |
> || WeakFréchet   | $\textcolor{red}{0.98281 (0.00025)}$    | $\textcolor{red}{0.93364 (0.00066)}$    | $\textcolor{red}{0.92836	(0.00075)}$    |
> || EditRealPenalty | $\textcolor{blue}{0.99830 (0.00007)}$   | $\textcolor{blue}{0.96695 (0.00030)}$   | $\textcolor{blue}{0.96573 (0.00035)}$   |
> || $m/4$-DTW       | $\textcolor{orange}{0.98949 (0.00016)}$ | $\textcolor{orange}{0.94757 (0.00048)}$ | $\textcolor{orange}{0.94475 (0.00057)}$ |
> |
> | TwoPersons | PartialDTW | ${0.94616 (0.00052)}$ | ${0.91961 (0.00026)}$ | ${0.92637 (0.00037)}$ |
> || WeakFréchet   | $0.95376 (0.00056)$         | $0.94832 (0.00002)$ | $0.95324 (0.00021)$         |
> || EditRealPenalty | $\textcolor{red}{0.85989 (0.00115)}$    | $\textcolor{red}{0.77931 (0.00004)}$    | $\textcolor{red}{0.76642 (0.00051)}$    |
> || $m/10$-DTW      | $\textcolor{blue}{0.96191 (0.00053)}$   | $\textcolor{blue}{0.94832 (0.00002)}$   | $\textcolor{blue}{0.95324 (0.00021)}$   |
>
> **Question 2:** More explicit parameter search over $k$.
>
> **Answer 2:** We refer to Table 2 in the appendix of our submission for a more extensive parameter search over $k$, cross-validated and independently repeated $100$ times. We may add the suggested evaluation in the next revision.

---

> > ### Comment · Reviewer_uWfM · 2025-04-01
> >
> > thanks for updated experiments with EditRealPenalties.  I retain my score of 4: accept.

---

### Official Review · Reviewer_MvAs · 2025-03-14

**Overall Recommendation:** 4

**Summary:**

This paper proposes a new distance measure called $k$-DTW, which is positioned as a interpolation between the classical DTW distance and the Fréchet distance. The technical novelty is to consider only the top $k$ matched distances in the alignment path, rather than summing all distances (as in DTW) or taking the maximum distance (as in Fréchet). The authors prove that $k$-DTW interpolates these two classical measures for $k = 1$ and $k$ sufficiently large. Empirical results show that $k$-DTW can yield stronger triangle-like properties than DTW, while still being more robust to outliers than Fr\'echet. A parametric search algorithm is introduced for computing $k$-DTW exactly (as well as a $(1+\varepsilon)$-approximation). The paper demonstrates applications in clustering (via hierarchical agglomerative clustering) and in nearest neighbor classification on synthetic and real-world datasets. The main contribution is that $k$-DTW addresses key shortcomings of Fréchet (outlier-sensitivity) and DTW (lack of metric-like properties) and, from experiments, appears beneficial in tasks like clustering and classification.

**Claims And Evidence:**

Claim 1: k-DTW is more robust to outliers than Fréchet distance.
Evidence: The synthetic-clustering experiment shows how spikes in type-A curves heavily penalize Fréchet but do less harm under k-DTW. However, the paper does not provide a standalone "robustness theorem." The authors rely primarily on intuitive arguments and empirical demonstrations.

Claim 2: k-DTW has stronger metric-like properties than DTW.
Evidence: They prove a relaxed triangle inequality with a factor of k, whereas DTW's violation can be worse by factors depending on curve length. The paper contains theoretical lemmas (especially Lemma 2.3) and the associated proofs. Empirically, the authors illustrate fewer pathological merges in clustering under k-DTW compared to DTW.

Claim 3: The proposed k-DTW algorithm is feasible for practical use.
Evidence: An exact algorithm and a (1+ε)-approximate algorithm are provided, though in worst cases it can be more expensive than classical DTW or Fréchet. The experiments show that it is slower than DTW and discrete Fréchet in practice.

**Essential References Not Discussed:**

I recommend referencing additional distances like partial DTW and weak Fréchet.

**Experimental Designs Or Analyses:**

- The authors constructed three curve types (A, B, C) to highlight the different shortcomings of Fréchet and DTW. This approach is sound: it clearly demonstrates how one or two spikes can dominate Fréchet, or how DTW can produce surprising alignments.
- They then used hierarchical clustering (single-linkage and complete-linkage) and visually showed the intra-/inter-cluster distances. The analysis effectively highlights differences among the measures.
- The real-world classification experiment (l-NN) is straightforward but adequate to underscore classification improvements. However, the paper could be enriched by including additional distance measures (partial DTW, weak Fréchet) for completeness.

**Methods And Evaluation Criteria:**

The authors' methods include:

- Defining k-DTW and demonstrating how to compute it via parametric search.
- Evaluating clustering via HAC with single-linkage and complete-linkage.
- Conducting l-nearest neighbor classification on real-world data.

These choices are appropriate to showcase the value of k-DTW in both supervised and unsupervised tasks. The design of synthetic data for highlighting robustness to peaks/outliers is quite clear. However, adding additional baseline distances like partial DTW or weak Fréchet could further solidify the empirical evaluations.

**Other Comments Or Suggestions:**

na

**Other Strengths And Weaknesses:**

Strengths:

- Proposes a simple and new bridging distance between two classic curve distances.
- Demonstrates some theoretical foundations like relaxed triangle inequality, dimension-free learning bounds, and provided practical benefits.
- Proposed an algorithm to compute k-DTW, plus the (1+ε)-approximation.

Weaknesses:

- No explicit "robustness theorem" about outliers, only partial arguments via examples.
- Time complexity is worse than classical DTW or discrete Fréchet distance, both in theory and experimental results.
- The experimental comparisons could be further supported by including partial DTW, weak Fréchet, or other established distances for completeness.

**Questions For Authors:**

You primarily demonstrate robustness to outliers through experiments. Could you formalize the observed outlier-robustness into a dedicated theorem or formal analysis?

Are there specific heuristic strategies or data-structural optimizations you foresee to further enhance the runtime efficiency of the k-DTW algorithm?

**Relation To Broader Scientific Literature:**

The paper explicitly compares $k$-DTW to the two classic distances: DTW and Fréchet. That is a direct relationship to established literature in computational geometry and time series analysis. We note that variants like weak Fréchet distance or partial DTW also exist, but they are not tested. The top-$k$ approach has precedents in top-$k$ optimization and the Ky-Fan norm (summing the $k$ largest singular values).I recommend referencing additional distances like partial DTW and weak Fr\'echet which might be beneficial to help the robustness comparison.

**Theoretical Claims:**

I checked the theoretical claims as laid out in Sections 2 and 3 of the paper:

- The proof of the relaxed triangle inequality (Lemma 2.3) is good.
- One point that is less formally addressed is the outlier-robustness: the paper does not contain a specialized "robustness theorem." Instead, it uses examples and partial discussions to argue that k-DTW dilutes spikes in the alignment cost.

---

> ### Author Rebuttal · Authors · 2025-03-31
>
> We thank the reviewer for the valuable feedback.
>
> **Question 1:** Formalize robustness.
>
> **Answer 1:** The concept of robustness for curve distance measures could be formalized as follows: given two curves whose matched vertices are at constant distance, say $1$, if we move one vertex away to increase the distance by a large value $\Delta$, then the average distance contributed per vertex increases through this modification by $\Delta/\Theta(k)$ for $k$-DTW, $k\in\{1,\ldots, m\}$. This means that Fréchet is largely dominated by the single outlier, while for $k$-DTW the increase averages out, so that up to a factor $(1\pm\varepsilon)$, one single large perturbation of order $\Delta \approx \varepsilon k$ is indistinguishable from tiny $\varepsilon$ perturbations of (all) single vertices. This supports and quantifies the intuition and may be included as a lemma in our next revision.
>
> To make the intuition more rigorous, we will examine the statistical breakdown point for $k$-DTW, which is defined to be the smallest fraction of vertices to be perturbed to corrupt the distance. A larger breakdown point indicates more robustness to arbitrary perturbations. As all $k$-DTW variants are closely related to the geometric median (minimizing a sum of distances), we will adapt the breakdown point analysis of the median [1] to our setting. It seems to be provable that the breakdown point of $k$-DTW is $\lfloor \frac{k+1}{2}\rfloor/m$. This would imply that as long as $k\in \omega(1)$, the breakdown point is asymptotically larger than for Fréchet and thus considerably more robust. However, to reach a constant breakdown point that provides robustness comparable to DTW for arbitrarily long curves, $k$ would still need to be of order $\Omega(m)$.
>
> [1] Lopuhaä, Rousseeuw - Breakdown Points of Affine Equivariant Estimators of Multivariate Location and Covariance Matrices, The Annals of Statistics, 1991
>
>
>
> **Question 2:** Heuristic strategies or data-structural optimizations to enhance the runtime.
>
> **Answer 2:** To enhance the runtime of our $k$-DTW algorithm *without* losing theoretical guarantees, we currently see two options:
>
> 1) The current DTW subroutine is the plain DP algorithm without any tweaks. It could potentially be enhanced using heuristic DTW speed-ups such as [2] as a black box.
>
> 2) Our first heuristic (line 206, right) could be enhanced by computing first only a subset of the DTW paths in a narrow band along the main diagonal. This works in linear $O(m)$ time and may be promising for reducing the variable 'mincost' early on so to cut off many unnecessary iterations.
>
> We will also add the challenging open problem of improving the top-$k$ optimization framework that our algorithms build upon. This will hopefully yield provably faster exact algorithms for other top-$k$ problems as well.
>
> [2] Silva, Batista - Speeding up all-pairwise dynamic time warping matrix calculation, ICDM 2016.
>
>
>
> **Question 3**: Additional baselines like partial DTW and weak Fréchet.
>
> **Answer 3**: Our claim is that $k$-DTW interpolates between the two extreme cases Fréchet and DTW. Comparisons to these two are thus most natural for supporting our claims. Please note that weak Fréchet has the same sensitivity to outlier vertices as standard Fréchet. Similarly, partial DTW admits only a factor $m$ triangle inequality. Hence, they do not contribute towards the goals of our paper.
>
> However, we agree on adding a few more baselines for completeness. We will thus include references, discussions and baseline experiments for partial DTW or weak Fréchet distance in addition to the suggestion of reviewer "uWfM". Please see our reply to "uWfM" for details and preliminary experiments.

---

### Decision · Program_Chairs · 2025-05-01

**Decision:**

Accept (poster)

**Comment:**

The paper introduces a new dissimilarity measure for polygonal curves called k-DTW, which improves on Dynamic Time Warping (DTW) by offering stronger metric properties and greater robustness to outliers. It provides both exact and approximate algorithms, theoretical learning bounds, and shows improved performance in clustering and nearest neighbor tasks. k-DTW also needs fewer samples than DTW.

All reviewers are positive about the paper. No major issues were raised, while minor ones were largely addressed in the rebuttal.

The finals scores are 3xAccept.

In addition to the theoretical contribution, the reviewers, overall, also appriciated the experiments (related to clustering and classification) and how they provided empirical evidence to support the authors' claims.

In the AC's view, this is a clear Accept.